# A Standardized Benchmark for Multilabel Antimicrobial Peptide Classification

**Sebastian Ojeda,**[*]   **Rafael Velasquez,**   **Nicolás Aparicio,**   **Juanita Puentes,**
**Paula Cárdenas**,   **Nicolás Andrade**,   **Gabriel González**,   **Sergio Rincón**,
**Carolina Muñoz-Camargo**,   **Pablo Arbeláez**

Universidad de los Andes, Colombia

## Abstract

Antimicrobial peptides have emerged as promising molecules to combat antimicrobial resistance. However, fragmented datasets, inconsistent annotations, and the lack of standardized benchmarks hinder computational approaches and slow down the discovery of new candidates. To address these challenges, we present the Expanded Standardized Collection for Antimicrobial Peptide Evaluation (**ESCAPE**), an experimental framework integrating over $80\,000$ peptides from 27 validated repositories. Our dataset separates antimicrobial peptides from negative sequences and incorporates their functional annotations into a biologically coherent multilabel hierarchy, capturing activities across antibacterial, antifungal, antiviral, and antiparasitic classes. Building on ESCAPE, we propose a transformer-based model that leverages sequence and structural information to predict multiple functional activities of peptides. Our method achieves up to a $2.56\%$ relative average improvement in mean Average Precision over the second-best method adapted for this task, establishing a new state-of-the-art multilabel peptide classification. ESCAPE provides a comprehensive and reproducible evaluation framework to advance AI-driven antimicrobial peptide research. [1]

## 1  Introduction

Antibiotics are crucial in modern medicine, enabling routine procedures and treating common infections. However, widespread misuse and overuse have led to the rise of antimicrobial resistance (AMR), where bacteria and other pathogens develop mechanisms to endure these drugs [1, 2]. This issue extends beyond bacteria, including viruses, fungi, and parasites that have rapidly evolved, hindering the treatment of infectious diseases globally [3]. The Institute of Health Metrics and Evaluation estimates that antimicrobial-resistant infections could cause over 39 million deaths between 2025 and 2050, with South Asia and Latin America facing the highest mortality rates [4]. Besides the impact of AMR on healthcare systems and population lifespan, it is also a concern for national economies across the globe [5]. A recent United Nations report warns that, without action on AMR, not only will healthcare costs increase, but also the global GDP may decrease by US$3.4 trillion and drive an additional 24 million people into extreme poverty by 2030 [6].

As a result, the scientific community has turned to alternative molecules capable of fighting infectious microorganisms without quickly triggering resistance. This process has led to the exploration of antimicrobial peptides (AMPs), which are either naturally occurring or synthetically designed proteins with a broad spectrum of antimicrobial properties [7]. Unlike traditional antibiotics, AMPs often act

---

[*]Corresponding author: `s.ojedaa@uniandes.edu.co`

[1]The ESCAPE Dataset is available at https://doi.org/10.7910/DVN/C69MCD and the ESCAPE Baseline code at https://github.com/BCV-Uniandes/ESCAPE.

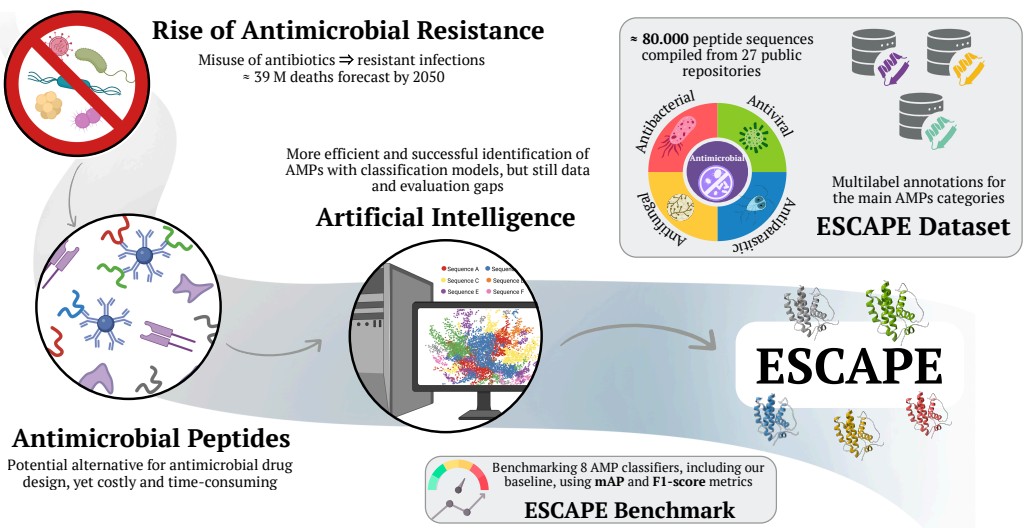

Figure 1: **Timeline of AMP Discovery and Computational Advances.** The rise of AMR underscores the urgent need for alternative therapies such as AMPs. While AI has shown promise in accelerating AMP discovery, progress is hindered by heterogeneous data and the absence of standardized evaluation protocols. We introduce ESCAPE to address these challenges and provide a robust foundation for future AI-driven methods.

through mechanisms harder for pathogens to evade, reducing the likelihood of resistance development [8]. Despite their therapeutic potential, the research and development of antimicrobial peptides, as with many pharmaceutical compounds, remains costly, time-consuming, and frequently unsuccessful. These difficulties are due to the inherent complexity of identifying effective AMPs and the extensive clinical trials and requirements novel drugs must undergo to reach the market and become profitable [9]. These challenges create a bottleneck that limits the widespread adoption of AMPs and highlights the need for more efficient and scalable discovery protocols.

To address the challenges of AMP discovery, researchers have explored using Artificial Intelligence (AI) tools to accelerate the identification of promising antimicrobial candidates [10]. Most existing models aim to predict whether a given peptide exhibits antimicrobial activity, often casting the task as a binary classification problem [11, 12, 13]. While this binary approach helps identify potentially active peptides, it disregards the proven ability of AMPs to interact with multiple types of microorganisms [14]. Despite this constraint, interest in AI-driven AMP classification tools has expanded in recent years. Some of the proposed methods include models based on Graph Neural Networks (GNNs) and other architectures inspired by advances in Natural Language Processing (NLP) [15, 16]. However, the performance of these models remains suboptimal, indicating a need for more effective modeling strategies and further exploration of architectures.

A significant limitation in the current literature is the inconsistency and insufficiency of data selected for training AI models [17]. Although the number of publicly available datasets continues to grow, most AI methods rely on individual datasets containing only a few hundred to a few thousand peptides [18]. Having such a limited number of examples from a single dataset when developing an AI model may restrict its learning potential and demonstrates the need for a more extensive set of data [19]. Furthermore, the absence of a standardized dataset or benchmarking framework makes it difficult to compare models reliably and to confidently identify which approach represents the state-of-the-art.

To overcome these limitations, we present three main contributions displayed in Fig. 1. First, we compile, curate, and standardize 27 public AMPs databases into the Expanded Standardized Collection for Antimicrobial Peptide Evaluation (ESCAPE), a comprehensive dataset containing over 80 000 peptides. We pre-process and validate all sequences to support robust AI-driven antimicrobial research. Second, we evaluate seven publicly available AMP classification methods on ESCAPE, adapting those designed initially for binary tasks to handle multilabel classification. To the best of our knowledge, this work results in the first benchmark that fairly compares existing approaches on a unified and scalable dataset for AMP multilabel classification. Finally, we introduce the ESCAPE

Baseline, a transformer-based architecture that leverages both sequence and structural information from peptides to predict not only whether it is antimicrobial, but also the types of pathogens it targets. Our baseline outperforms the state-of-the-art methods on the complete ESCAPE Dataset with a relative average improvement with respect to the second-best method of 2.56% and 1.90% in mean Average Precision (mAP) and F1-score, respectively.

## 2 Related Work

Given the global concern for antimicrobial resistance, there has been a noticeable rise in the development of AMP databases in recent years. Building on these resources, researchers have developed numerous AI models to analyze and identify potential AMP candidates to optimize antimicrobial drug discovery.

### 2.1 AMP Databases

In the search for novel AMPs, researchers have developed several databases, each containing peptides annotated with diverse biological activities [20]. These databases can be broadly categorized into general [21, 22, 23, 24] and specialized databases [25, 26, 27]. General databases span a wide range of peptide functions or classes. In contrast, specialized databases focus on narrower aspects, such as peptide origin [26] or the type of target organism [25], [27]. The number of peptide entries and the granularity of functional classes vary substantially among databases. For instance, dbAMP [21] comprises 33 065 peptides annotated with 58 distinct functional classes, while DRAMP [22] includes 30 260 entries but only 8 classes. Moreover, the hierarchical level of these classes is often inconsistent across databases. Namely, LAMP2 [23] annotates 23 253 peptides into 38 classes, including "anti-Gram negative" and "anti-Gram positive," which researchers can interpret as subclasses of a broader "Antibacterial" category. In contrast, SATPdb [24] assigns its 19 192 peptides to just 10 classes, among them a single "Antibacterial" label. Hence, currently available AMP datasets present key limitations. The class granularity and hierarchy discrepancies complicate training predictive models and comparing their performance across datasets.

Additionally, training models on datasets composed entirely of peptides obtained using a single experimental technology may introduce methodological biases, ultimately hampering the model's ability to generalize across broader peptide sources [28]. The ESCAPE Database addresses these limitations by integrating a wide range of public AMP datasets with more than 80 000 peptides obtained from various sources. In addition, we standardize the class annotation system by curating a concise and biologically meaningful hierarchy of antimicrobial functions, thus improving the interpretability of annotations and facilitating dataset integration.

### 2.2 AMP Benchmarks

Benchmarking is critical in evaluating AMP prediction models, yet standardized protocols for consistent comparison are still missing. Many studies rely on custom datasets and train-test splits without releasing exact partitions, hindering reproducibility and fair comparison against other methods [29]. Furthermore, most benchmarking efforts focus on binary classification tasks that fail to capture the functional diversity and therapeutic relevance of AMPs [30]. Although several studies have introduced multilabel classification approaches for AMPs [15, 31], the field still lacks a standardized, openly accessible benchmark designed to support rigorous evaluation in multilabel antimicrobial peptide prediction. In this context, the ESCAPE Benchmark is a significant advancement by enabling rigorous evaluation of seven state-of-the-art models on the multilabel AMP classification task, thereby facilitating fair and transparent comparisons of AI methodologies on a large-scale peptide dataset.

### 2.3 AMP Classification Models

There are two main categories of AMP classification methods in the current literature: those that rely exclusively on raw amino acid sequences and those that incorporate bioinformatically derived descriptors using specialized libraries. Sequence-focused methods include AMPlify [11], which employs a bidirectional LSTM with multi-head and context attention to generate a summary vector. Other methods use Large Language Models (LLMs) [32] as the backbone of the architecture. For example, TransImbAMP [31] uses a BERT model pretrained in a self-supervised manner via masked token

prediction and fine-tunes a fully connected layer attached to its outputs for the AMP classification task. AMP-BERT [12], another sequence-based model, also employs a pretrained BERT, but with an inserted class token whose embedding guides the classifier. More recently, dsAMPGAN [33] integrates CNN, Attention, and BiLSTM layers with transfer learning to perform AMP classification and function prediction, while AMPpred-DLFF [34] combines ESM-2 [35] embeddings with graph attention networks and CNN modules to capture both spatial and sequential information.

In contrast, feature-augmented approaches compute additional descriptors before modeling. For instance, amPEPpy [13] feeds CTD features (composition, transition, distribution of physicochemical amino-acid classes) to a Random Forest, and AMPs-Net [15] represents peptides as graphs enriched with physicochemical node and edge attributes to then process the peptides with a GNN. PEP-Net [36] fuses one-hot amino-acid identities, computed physicochemical properties, and high-dimensional protein language model embeddings through residual convolutional and Transformer blocks to capture local information and global contextual information. AVP-IFT [37] employs a dual-branch framework integrating contrastive learning with a transformer network enhanced with biophysical and chemical properties. Recent work has also introduced ensemble-based feature-augmented approaches, including StackAMP [38], AMP-RNNpro [39], and StackPIP [40], which combine diverse peptide descriptors with multiple machine learning models to improve classification performance. Unlike prior methods, the ESCAPE Baseline introduces a bidirectional cross-attention mechanism by integrating the peptide sequence and its 3D distance representation, enabling the combination of spatial structural information with the sequence and leading to superior performance.

# 3 Expanded Standardized Collection for Antimicrobial Peptide Evaluation

Current research on AMP prediction faces a critical bottleneck due to fragmented, inconsistent, and small-scale datasets that vary widely in format, annotation standards, and functional coverage [17]. These limitations hinder the development of robust predictive models and complicate fair comparisons across methods. To overcome this gap, we introduce the ESCAPE Dataset, a unified collection of antimicrobial peptides compiled from 27 public repositories. This multilabel framework standardizes functional annotations to reflect the biological taxonomy of infectious agents, resulting in five classes: four main AMP activities (antibacterial, antifungal, antiviral and antiparasitic) and a fifth category (antimicrobial) that represents the general ability to act on any microorganism. Peptides that do not exhibit antimicrobial properties, and thus do not belong to any of the five classes, are considered Non-AMPs.

## 3.1 Data Compilation

To build the ESCAPE Dataset, we collect experimentally validated and manually curated antimicrobial peptide entries from 27 public databases: BIOPEP-UWM Database [41], CPPsite 2.0 [42], CAMPR3 [43], TumorHoPe [44], APD3 [45], SPdb [46], ParaPep [47], CancerPPD [27], BrainPreps [48], Quorumpeps [49], YADAMP [25], LAMP2 [23], Milkampdb [50], DADP [26], AntiTbPdb [51], PeptideDB [52], NeuroPrep [53], SATPdb [24], BioDADPep [54], NeuroPedia [55], DFBP [56], dbAMP 3.0 [21], DRAMP 4.0 [22], AVPdb [57], Hemolytik [58], DBAASP v3 [59], and UniProt [60]. Each source contributes unique peptide profiles across four functional classes: antibacterial, antifungal, antiviral, and antiparasitic. This structure encapsulates key differences in mechanisms of action, such as disarrangement of bacterial membranes [61], inhibition of cell wall biosynthesis [62], and interference with viral assembly [57], among others.

For the integration of Non-AMP samples, we follow the methodology outlined in TransImbAMP [31], focusing on selecting non-antimicrobial sequences from UniProt [60]. We apply stringent exclusion criteria, removing entries associated with keywords such as "membrane," "toxic," "secretory," "defensive," "antibiotic," "anticancer," "antiviral," or "antifungal". To expand this set, we incorporate peptides from curated datasets known for their non-antimicrobial functions [54, 47]. This dual strategy of exclusion-based filtering and targeted selection constructs a high-confidence negative set that effectively distinguishes non-AMP sequences, providing a robust contrast for supervised training. Supplementary Material Section A offers specific details on the creation of the dataset, regarding handling the compiled databases and their associated licenses.

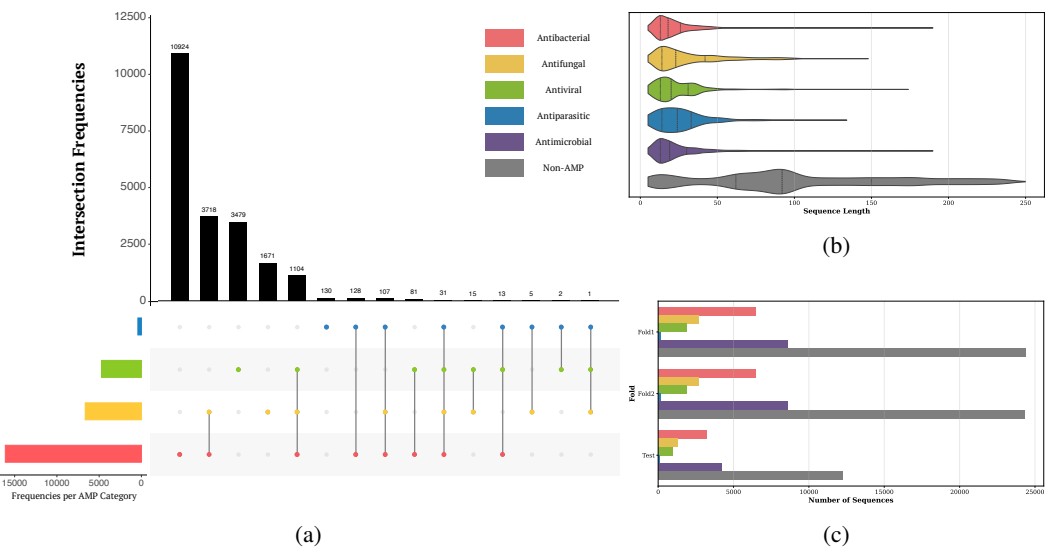

Figure 2: **Overview of ESCAPE Dataset Composition and Statistics.** (a) Multilabel distribution of AMPs across the four functional classes in ESCAPE Dataset, (b) Sequence length distribution for AMPs and non-AMPs, and (c) Distribution of AMP and non-AMP sequences in the two folds and the test set of the dataset.

## 3.2 Data Processing and Cleaning

We remove sequences containing synthetic residues such as pyrrolysine (O), selenocysteine (U), $\beta$-alanine (Bal), 3-naphthylalanine (Nal), or 2-aminobutanoic acid (Abu), following the pipeline that AMP-Net [15] proposes. We exclude entries with undefined amino acids (X) and retain degenerate codes J, B, and Z, treating them as biologically valid representations of leucine/isoleucine, aspartic acid/asparagine, and glutamic acid/glutamine, respectively. We also enforce length constraints, retaining only peptides with lengths between 5 and 250 residues, ensuring structural relevance and alignment with established peptide standards [15]. Finally, to mitigate redundancy, we consolidate duplicate sequences across repositories and integrate their corresponding functional annotations into a unified multilabel vector.

## 3.3 ESCAPE Dataset

The ESCAPE Dataset comprises $60\,950$ non-AMPs and $21\,409$ AMPs, which are functionally annotated into four major antimicrobial categories: antibacterial, antifungal, antiviral, and antiparasitic. Figure 2a shows that most AMPs ($16\,106$) exhibit antibacterial activity, and $10\,924$ belong exclusively to this class. The most common combination is antibacterial and antifungal sequences (4960 peptides), while only 1671 peptides are solely antifungal. Antiviral entries total 4726 (3479 unique), and antiparasitic peptides number 417 (130 unique). Rarer multilabel groups include primarily antiparasitic sequences exhibiting antifungal or antiviral activity.

Fig. 2b presents the sequence length distribution for AMPs and non-AMPs. AMP sequences are predominantly centered around 30 amino acids, which is characteristic of this type of peptide structure [63]. In contrast, non-AMPs exhibit a broader length range, with an average of 90 amino acids, and most sequences fall between 50 and 100 residues. This distinction highlights the inherent structural differences between AMPs and non-AMP sequences.

The ESCAPE Dataset is divided into three folds: two used for cross-validation and one reserved for independent testing. This partitioning strategy ensures comprehensive model evaluation by maintaining equal label distributions across folds, as shown in Fig. 2c. This consistency supports reliable performance assessment and minimizes overfitting risks between functional classes.

### 3.4 ESCAPE Benchmark

To evaluate the performance of existing AMP classification methods under the standardized multilabel framework introduced by ESCAPE, we benchmark seven representative models with publicly available implementations that allow reproducibility: AMPlify [11], AMP-BERT [12], TransImbAMP [31], amPEPpy [13], AMPs-Net [15], PEP-Net [36], and AVP-IFT [37]. We adapt each model to support multilabel classification and we train it with a two-fold cross-validation scheme. In Supplementary Section B, we provide per-model implementation details, with the training hyperparameters summarized in Supplementary Table 2. To establish robust and consistent evaluation, we train each method three times with the random seeds (42, 1665, 8914) across all models. We perform the final evaluation with an ensemble that averages the probabilities from the two trained models. We assess performance using mean Average Precision (mAP) and F1-score, two widely used metrics for multilabel settings with severe class imbalance typical of AMP prediction [64]. We report the mean and standard deviation across the three runs.

## 4  ESCAPE Baseline

Given the limitations of existing models in addressing multiclass classification of antimicrobial peptides, we design a method that combines sequential and structural information to perform the classification task on the ESCAPE Database. We introduce the ESCAPE Baseline model, a transformer-based architecture that integrates sequence and structural modalities through bidirectional cross-attention, and we provide a detailed account of its design and implementation.

### 4.1  Input Peptide Representation

**Sequence Representation.** Let $\mathcal{S}$ denote the set of all peptide sequences. Each sequence $s \in \mathcal{S}$ is defined as an ordered list of amino acids $s = [a_1, a_2, \ldots, a_N]$ with $N$ residues, where each $a_i \in \mathcal{A}$. The vocabulary $\mathcal{A}$ consists of 26 amino acid symbols (including rare and ambiguous codes), together with a special padding token, resulting in a vocabulary length of 27 [11]. Let $\mathcal{T} = \{t_1, t_2, \ldots, t_{27}\}$ be a finite set of discrete tokens, with $|\mathcal{T}| = 27$, representing the token vocabulary. We define a bijective mapping $f : \mathcal{A} \to \mathcal{T}$ such that for every $a_i \in \mathcal{A}$ there exists a unique token $t_i = f(a_i) \in \mathcal{T}$. This one-to-one correspondence allows each sequence $s$ to be equivalently represented as a token sequence $t = [t_1, t_2, \ldots, t_{\mathcal{L}}]$ over the vocabulary $\mathcal{T}$. All sequences are either truncated or zero-padded to a fixed length $\mathcal{L}$.

**Structural Representation.** For each peptide, structural information is collected from the UniProt [60] and Protein Data Bank (PDB) [65] repositories. When experimental structures are unavailable, we use RosettaFold [66] and AlphaFold3 [67] to predict the three-dimensional conformations. These models are state-of-the-art deep learning methods for protein structure prediction. Given a peptide with $N$ amino acids, we compute a distance matrix $\mathcal{M} \in \mathbb{R}^{N \times N}$, based on the 3D structure. Each element $\mathcal{M}_{i,j}$ corresponds to the Euclidean distance between the C$\alpha$ atoms of residues $i$ and $j$: $\mathcal{M}_{i,j} = \|\mathbf{r}_i - \mathbf{r}_j\|$, where $\mathbf{r}_{i,j} \in \mathbb{R}^3$ denotes the spatial coordinates of the $i^{\text{th}}$, $j^{\text{th}}$ residue. To ensure compatibility across peptides of varying lengths, $\mathcal{M}$ is resized to a fixed dimension of $224 \times 224$, enabling uniform input to the structural encoder.

### 4.2  Model Architecture

The ESCAPE Baseline model is built upon a dual-branch transformer architecture designed to jointly encode the sequence and structural modalities of peptides, as illustrated in Fig. 3. Each branch independently processes one modality using a specific transformer encoder, and the resulting representations are fused via a bidirectional cross-attention mechanism to enable cross-domain interaction.

**Sequence Module.** The sequence branch takes the tokenized peptide sequence and maps each token to a 256-dimensional vector using a learnable embedding matrix. In addition, we include a special [CLS] token to capture global sequence-level information and add positional embeddings to preserve the order of amino acids. Later, this feeds the resulting embeddings through a stack of $\mathcal{D}$ Transformer encoder layers. This setup enables the model to learn local and long-range dependencies within the

sequence. The [CLS] token at the output serves as a compact representation of the peptide's primary structure.

**Structure Module.** The structural branch receives as input a single-channel $224 \times 224$ distance matrix $\mathcal{M}$, where each entry indicates the Euclidean distance between a pair of $C\alpha$ atoms in the peptide's 3D structure. A 2D convolution with kernel size and stride of 16 partitions this matrix into non-overlapping $16 \times 16$ patches, producing a grid of flattened patches. The model projects each patch into a 192-dimensional embedding, creating a sequence of patch embeddings. The model adds a learnable [CLS] token at the beginning of the sequence to aggregate global structural information and appends fixed positional encodings to preserve spatial relationships. The model processes the sequence through a stack of $\mathcal{D}$ Transformer encoder layers that capture local and long-range spatial dependencies. The structure branch outputs the [CLS] token, which captures a compact representation of the peptide's 3D conformation.

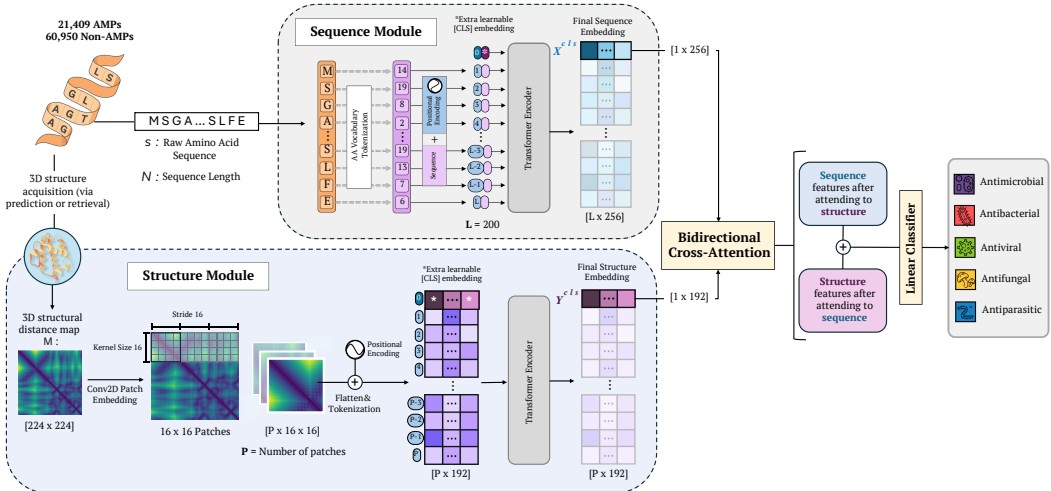

Figure 3: **ESCAPE Baseline Architecture Overview.** The model encodes each peptide using two parallel branches: the sequence module tokenizes amino acid residues. It extracts a [CLS] representation through a Transformer encoder. In contrast, the structure module processes a $224 \times 224$ distance matrix by embedding non-overlapping patches and applying a Transformer stack to produce a structural [CLS] token. A bidirectional cross-attention mechanism fuses these two representations by allowing each modality to attend to the other. The model concatenates the resulting attended CLS vectors and passes them through a linear layer to generate the final multilabel prediction vector.

**Bidirectional Cross-Attention.** We apply a bidirectional cross-attention mechanism to integrate information from sequence and structure modalities after independently encoding each branch. As detailed in the Sequence and Structure Modules, our model processes the amino acid sequence and the distance matrix separately through their transformer encoders. Each encoder produces a contextual embedding matrix and a corresponding [CLS] token that summarizes global information.

Let $\mathbf{X} \in \mathbb{R}^{\mathcal{L} \times 256}$ and $\mathbf{Y} \in \mathbb{R}^{\mathcal{P} \times 192}$ denote the sequence and structure embedding matrices, respectively, where $\mathcal{L}$ and $\mathcal{P}$ are the modality-specific lengths, including the [CLS] token. The corresponding [CLS] embeddings are denoted as $\mathbf{X}_{\text{cls}}$ and $\mathbf{Y}_{\text{cls}}$.

To enable cross-modal interaction, we apply attention in both directions. First, the sequence attends to the structural features where $\mathbf{Q}_x = \mathbf{X}\mathbf{W}_Q^x$ are the queries derived from the sequence, and $\mathbf{K}_y = \mathbf{Y}\mathbf{W}_K^y$, $\mathbf{V}_y = \mathbf{Y}\mathbf{W}_V^y$ are the keys and values projected from the structure branch. The attention output $\mathbf{A}_x$ is added to the original sequence embeddings via a residual connection and refined through a feedforward network. Conversely, the structural representation attends to the sequence where $\mathbf{Q}_y = \mathbf{Y}\mathbf{W}_Q^y$, and $\mathbf{K}_x = \mathbf{X}\mathbf{W}_K^x$, $\mathbf{V}_x = \mathbf{X}\mathbf{W}_V^x$ are the projections from the sequence encoder.

$$\mathbf{A}_x = \text{softmax}\left(\frac{\mathbf{Q}_x\mathbf{K}_y^\top}{\sqrt{d}}\right)\mathbf{V}_y, \qquad \mathbf{A}_y = \text{softmax}\left(\frac{\mathbf{Q}_y\mathbf{K}_x^\top}{\sqrt{d}}\right)\mathbf{V}_x,$$

The bidirectional attention mechanism enables each modality to gather contextual information from the other by focusing on the most informative regions of the complementary representation. After bidirectional cross-attention, we concatenate the updated [CLS] from both modalities and pass the resulting vector through a classification head to produce the final prediction.

### 4.3 Implementation Details

We train the ESCAPE Baseline on a NVIDIA GPU A100 with a batch size $64$ for $100$ epochs, using the AdamW optimizer and a learning rate of $1 \times 10^{-4}$. We incorporate dropout layers within each transformer encoder block to mitigate overfitting and enhance generalization. In the sequence branch, we tokenize each input peptide and pad it to a fixed length of $\mathcal{S} = 200$, embedding each amino acid into a 256-dimensional space. The sequence encoder comprises $4$ transformer layers, each with $8$ attention heads. In the structural branch, we represent each peptide's distance matrix as a $224 \times 224$ single-channel image and divide it into non-overlapping $16 \times 16$ patches. We flatten each patch and project it into a 192-dimensional embedding. We then process the resulting patch embeddings with a transformer encoder that shares the same configuration as the sequence branch, consisting of $4$ layers and $8$ attention heads, to capture spatial and geometric dependencies within the peptide structure.

## 5 Experiments and Discussion

### 5.1 Main Results

Table 1 reports the mean and standard deviations over the three random seeds of the overall and per-class F1-scores, while Table 2 presents the corresponding mAP values. The results demonstrate that our baseline consistently surpasses all seven state-of-the-art methods across both metrics, reaching relative improvements of $1.90\%$ in overall F1-score and $2.56\%$ in mAP compared to the second-best method.

The per-class evaluation reveals that ESCAPE Baseline achieves the most substantial gains in the least represented categories. In particular, it increases the AP for the antiparasitic class by $35.7\%$ relative to the second-best method. Among sequence-based models, AMPlify attains the highest performance, with an average F1-score of $68.5\%$ and mAP of $70.3\%$. AVP-IFT follows closely, achieving $66.5\%$ F1 and $68.8\%$ mAP by integrating physicochemical descriptors through a feature-augmented design. These comparisons indicate that no single modeling strategy, whether sequence-focused or feature-augmented, consistently dominates the others.

Table 1: **Overall and Per-Class F1-Scores on the ESCAPE Benchmark.** F1-scores for each model averaged over the 42, 1665, and 8914 random seeds on the 5-class multilabel classification task in the ESCAPE Benchmark (%).

| Method | F1-Score | Antibacterial | Antiviral | Antifungal | Antiparasitic | Antimicrobial |
|---|---|---|---|---|---|---|
| AMPs-Net [15] | $57.7 \pm 0.70$ | $78.9 \pm 0.77$ | $59.2 \pm 0.79$ | $61.1 \pm 0.51$ | $5.9 \pm 0.71$ | $83.5 \pm 0.79$ |
| TranslmbAMP [31] | $62.0 \pm 0.70$ | $87.1 \pm 0.96$ | $59.2 \pm 0.50$ | $54.7 \pm 0.51$ | $21.8 \pm 0.81$ | $87.2 \pm 0.75$ |
| AMP-BERT [12] | $64.7 \pm 0.64$ | $89.3 \pm 0.27$ | $63.0 \pm 0.95$ | $60.2 \pm 0.26$ | $20.6 \pm 3.52$ | $90.5 \pm 0.22$ |
| PEP-Net [36] | $65.5 \pm 0.61$ | $\mathbf{89.5 \pm 0.10}$ | $58.1 \pm 0.78$ | $\mathbf{65.2 \pm 0.55}$ | $22.8 \pm 0.61$ | $\mathbf{91.2 \pm 0.15}$ |
| amPEPpy [13] | $66.5 \pm 0.37$ | $87.6 \pm 0.07$ | $61.6 \pm 2.02$ | $60.4 \pm 1.90$ | $34.7 \pm 0.98$ | $90.9 \pm 3.78$ |
| AVP-IFT [37] | $66.5 \pm 0.59$ | $89.1 \pm 0.47$ | $\mathbf{64.8 \pm 0.06}$ | $60.7 \pm 0.55$ | $28.0 \pm 3.84$ | $89.9 \pm 0.31$ |
| AMPlify [11] | $68.5 \pm 0.77$ | $88.8 \pm 0.26$ | $60.0 \pm 1.05$ | $65.0 \pm 1.57$ | $40.9 \pm 2.48$ | $90.0 \pm 0.30$ |
| **ESCAPE Baseline (Ours)** | $\mathbf{69.8 \pm 0.43}$ | $88.8 \pm 0.34$ | $64.4 \pm 0.88$ | $61.0 \pm 0.75$ | $\mathbf{44.8 \pm 0.50}$ | $90.0 \pm 0.32$ |

Overall, performance trends highlight that the effectiveness of the model depends on the synergy between input representations and architectural design, rather than on the mere inclusion of additional descriptors. By combining sequence and structural modalities, ESCAPE Baseline leverages richer biological representations to improve generalization. The architecture also supports flexible operation under different configurations using only sequence information, only 3D structural data, or both, achieving its best performance when integrating the two. This versatility comes from its higher capacity and ability to align complementary modalities during training, establishing ESCAPE Baseline as both a robust and adaptable framework for multilabel AMP classification.

Furthermore, as shown in Figure 2 of the Supplementary Material, model size does not exhibit a consistent relationship with predictive performance across the evaluated methods. Notably, although

the top-performing model has nearly 9 million parameters, the second-best model overall is based on a Random Forest classifier, making it the least computationally demanding method in our benchmark. Conversely, as Table 1 and Table 2 report, BERT-based approaches do not rank among the top three performing models. These findings suggest that more complex architectures do not necessarily yield superior results for multilabel AMP classification. Additionally, our results underscore the limitations of large language models when applied to domains outside of natural language. Despite the domain adaptation efforts through BERT fine-tuning in models such as TransImbAMP and AMP-BERT, these methods fail to fully accommodate the peptide-specific data and, as a result, underperform in this task.

Table 2: **Mean and Per-Class AP Results on the ESCAPE Benchmark.** AP for each model averaged over the 42, 1665, and 8914 random seeds on the 5-class multilabel classification task in the ESCAPE Benchmark (%).

| Method | mAP | Antibacterial | Antiviral | Antifungal | Antiparasitic | Antimicrobial |
|---|---|---|---|---|---|---|
| AMPs-Net [15] | $54.6 \pm 0.86$ | $82.5 \pm 0.72$ | $51.2 \pm 0.88$ | $53.1 \pm 0.84$ | $5.3 \pm 0.67$ | $82.1 \pm 0.80$ |
| TransImbAMP [31] | $64.9 \pm 1.11$ | $92.5 \pm 1.23$ | $65.0 \pm 1.63$ | $56.3 \pm 0.96$ | $16.7 \pm 0.86$ | $94.0 \pm 0.90$ |
| AMP-BERT [12] | $66.9 \pm 1.17$ | $92.3 \pm 0.59$ | $65.9 \pm 1.84$ | $61.5 \pm 2.28$ | $21.4 \pm 2.61$ | $93.6 \pm 1.25$ |
| amPEPpy [13] | $68.5 \pm 0.48$ | $93.9 \pm 0.24$ | $67.7 \pm 0.28$ | $62.2 \pm 0.27$ | $23.8 \pm 1.61$ | $95.2 \pm 0.05$ |
| PEP-Net [36] | $68.4 \pm 0.53$ | $\mathbf{95.2 \pm 0.21}$ | $61.2 \pm 0.67$ | $\mathbf{72.6 \pm 0.78}$ | $16.2 \pm 0.84$ | $\mathbf{96.7 \pm 0.26}$ |
| AVP-IFT [37] | $68.8 \pm 0.50$ | $94.3 \pm 0.49$ | $\mathbf{71.1 \pm 0.36}$ | $63.3 \pm 1.36$ | $20.0 \pm 4.25$ | $95.5 \pm 0.50$ |
| AMPlify [11] | $70.3 \pm 0.87$ | $94.0 \pm 0.19$ | $66.1 \pm 5.56$ | $68.3 \pm 4.27$ | $27.7 \pm 1.33$ | $95.3 \pm 0.31$ |
| **ESCAPE Baseline (Ours)** | $\mathbf{72.1 \pm 0.60}$ | $94.2 \pm 0.21$ | $69.8 \pm 0.46$ | $63.4 \pm 0.74$ | $\mathbf{37.6 \pm 2.87}$ | $95.6 \pm 0.04$ |

Per-class analysis reveals substantial variation in predictive performance across functional categories. As the number of samples per class decreases, most models exhibit a proportional decline in both mAP and F1-score, independent of their underlying architecture or feature design. The antiparasitic and antiviral classes, which contain the fewest examples, yield the lowest scores across all evaluated methods, highlighting the intrinsic difficulty of learning under severe data scarcity. In contrast, categories with broader representation, such as antibacterial and antifungal, display more stable results and narrower variability among models. This trend underscores the impact of label imbalance as the dominant factor shaping overall performance, suggesting that future improvements should focus on better representation learning rather than on increasing model complexity alone.

## 5.2 Ablation Experiments

To assess the individual contribution of each peptide representation modality in our baseline, we conduct an ablation experiment using the 42 seed. We evaluate three configurations: one using only the sequence module, another using only the structural module based on distance matrices, and a third combining both through the cross-attention module. Table 3 shows that the sequence representation provides considerably more informative features than the structural one: with sequence-only input, our model achieves 21.7% higher mAP and 20.7% higher F1 score than with the distance matrix alone. This gap likely arises because the structural view captures spatial arrangement but omits explicit biochemical identities, thereby limiting the model's ability to exploit residue-level patterns critical for antimicrobial activity. These findings highlight that the biological composition of the peptide encoded in the sequence plays a decisive role in classification performance. Yet Table 3 also shows that combining both representations via cross-attention yields the best overall results: while the sequence-only variant is already strong, adding structural cues provides a complementary signal that further improves prediction quality.

Table 3: **ESCAPE Baseline Ablation Experiments.** mAP (%) and Overall F1-score (%) reported for the 42 random seed trained model.

| Structure Module | Sequence Module | Cross Attention | mAP | F1 |
|---|---|---|---|---|
| ✓ | · | · | 47.7 | 46.9 |
| · | ✓ | · | 69.4 | 67.6 |
| ✓ | ✓ | ✓ | **72.7** | **69.5** |

As detailed in Section 4, we obtain structural information for most peptides from UniProt and PDB, and infer the remaining structures using public generative models [66, 67]. We also run a sensitivity experiment to assess the impact of using predicted structures. Supplementary Section C

(Table 5) shows that relying solely on predicted structures leads to reduced performance compared to experimental structures, with absolute drops of 1.5% in mAP and 1.9% in F1. These results suggest that artificially generated structures may introduce additional sources of error inherent to those models, potentially degrading the quality of the structural data and impairing the model's ability to accurately classify peptides.

## 5.3 Limitations and Broader Impact

In this work, we present a carefully curated and extensive dataset for AMP discovery. An important limitation arises from the scope of the domain, as the diversity of peptides in nature is vast and it is not feasible to capture all existing variants or ensure that our dataset fully represents the underlying distribution. Nonetheless, our benchmark provides a standardized and transparent framework for evaluating AMP classification models, helping to identify methodological gaps and guide future improvements. By promoting reproducibility and comparative analysis in this area, our work contributes to advancing computational tools for AMP discovery, which may support efforts in global health, particularly in the context of antibiotic resistance. However, further validation in real-world biological and clinical settings is required before deployment of such models.

Another limitation arises from the inherent differences in sequence length distributions. Antimicrobial peptides are naturally shorter than most non-AMPs, a characteristic tied to their biological function. In constructing ESCAPE, we aim to preserve this molecular distinction while avoiding strong correlations that could bias classification. The resulting dataset maintains realistic differences between classes without allowing sequence length to dominate predictive performance, supporting a fairer evaluation of computational models.

## 5.4 Ethical Considerations

From an ethical standpoint, ESCAPE is constructed entirely from publicly available, experimentally validated datasets, each distributed under its respective license. While the benchmark provides a foundation for advancing computational methods in antimicrobial peptide research, our contribution remains focused on methodological innovation rather than direct therapeutic or drug design applications.

At the same time, we acknowledge the potential risks arising from irresponsible use of this resource. Models trained on ESCAPE could, if misused, be employed to generate peptides without appropriate experimental validation, raising concerns about toxicity or biosecurity. To mitigate such risks, we encourage the research community to operate within established ethical, biosafety, and regulatory standards and to ensure that all experimental and computational findings are reported transparently and with accountability.

# 6 Conclusions

We introduce **ESCAPE**, the first standardized benchmark for multilabel antimicrobial peptide classification, designed to overcome key limitations of existing resources, including data fragmentation, inconsistent annotations, and limited functional scope. ESCAPE integrates over 80.000 peptides from 27 curated repositories into a biologically grounded multilabel framework encompassing antibacterial, antifungal, antiviral, antiparasitic and antimicrobial classes. It also includes a rigorously filtered set of non-antimicrobial sequences to support reliable supervised training and better reflect real-world prediction settings. Building upon this foundation, we propose a baseline using a transformer-based architecture that leverages sequence and structural information. We demonstrate that ESCAPE Benchmark enables fair and reproducible comparison across models and functional classes, setting a new standard for AI-driven AMP discovery, particularly in underrepresented categories such as antiviral and antiparasitic.

# 7 Acknowledgements

This research was partially funded by the Colombian Ministry of Science, Technology, and Innovation (Minciencias), under Cod. 1204-937-101846, CR 19576-2024 Call for Fundamental Research. This work was supported by Azure sponsorship credits granted by Microsoft's AI for Good Research Lab.

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
