# Supplementary Material
# A Standardized Benchmark for Multilabel Antimicrobial Peptide Classification

**Sebastian Ojeda,*** **Rafael Velasquez,** **Nicolás Aparicio,** **Juanita Puentes,** **Paula Cárdenas**, **Nicolás Andrade**, **Gabriel González**, **Sergio Rincón**, **Carolina Muñoz-Camargo**, **Pablo Arbeláez**

Universidad de los Andes, Colombia

## A  ESCAPE Dataset Compilation

### A.1  Compilation and Standardization of Datasets

We compile ESCAPE from 27 peptide databases by systematically extracting experimentally validated antimicrobial peptides annotated for antibacterial, antifungal, antiparasitic, or antiviral activity. Databases exclusively focusing on a single category, such as AVPdb [1] (antiviral), are directly mapped to one of the four target classes. For negative examples, we filter peptides from external sources unrelated to antimicrobial activity, such as anticancer (e.g., CancerPPD [2], TumorHoPe[3]) and neuroactive peptide databases (e.g., NeuroPep [4], BrainPeps [5]). Additionally, we follow the methodology outlined in TransImbAMP[6], selecting non-antimicrobial peptides from UniProt [7] by applying strict exclusion criteria. Specifically, we discard sequences containing keywords such as "membrane," "toxic," "secretory," "defensive," "antibiotic," "anticancer," "antiviral," or "antifungal" to enhance the quality of the negative class.

For large and hierarchically structured databases such as DBAASP[8], DRAMP[9], dbAMP (with species-level annotations)[10], and SATPdb (which lists 38 functional categories)[11], we retain all peptides with annotations that map either directly or through hierarchical or taxonomic relationships to one of our four defined antimicrobial classes (antibacterial, antifungal, antiparasitic, antiviral). This includes entries annotated at the level of function (e.g., "antifungal"), target phenotype (e.g., "anti-Gram positive"), or biological taxonomy (e.g., species or genus) when these map to our label taxonomy. We exclude peptides whose annotations lack any functional or taxonomic correspondence to our classes. When databases separate targets by phenotype, such as "anti-Gram positive" and "anti-Gram negative," we merge these into a unified antibacterial class. We also perform manual curation for complex hierarchical annotations, consolidating entries from species, family, or domain levels under the most appropriate class. After this initial selection, we identify and resolve duplicate sequences across datasets by merging complementary annotations. For instance, if a peptide appears in multiple sources with evidence of both antifungal and antibacterial activity, we retain a single entry enriched with both labels.

### A.2  Licenses and copyright

To build the ESCAPE benchmark, we aggregate data from 27 publicly available peptide databases covering antibacterial, antifungal, antiparasitic, and antiviral peptides, as well as peptides with no known antimicrobial activity. Table 1 lists the number of peptides available in each source along with their corresponding license terms. We apply a rigorous filtering and selection pipeline to construct a legally compliant benchmark. For datasets under permissive licenses (e.g., CC BY or CC BY-NC),

---

*Corresponding author: s.ojedaa@uniandes.edu.co

we include the relevant entries directly in ESCAPE. When licenses restrict redistribution (e.g., Oxford University Press[12][13][14], Elsevier[15], Springer Nature[5]), we exclude the raw data and instead reference hashed identifiers and provide scripts to enable reproducibility. This strategy ensures that ESCAPE adheres to academic licensing standards while offering broad coverage of experimentally validated antimicrobial and non-antimicrobial peptides.

Table 1: **Overview of peptide databases integrated into the ESCAPE benchmark**. Here we detail the number of peptides and associated licensing terms for each source.

| Database Name | Number of Peptides | License |
|---|---|---|
| BIOPEP-UWM Database [16] | 3634 | CC BY 4.0 |
| CPPsite 2.0 [17] | 1155 | CC BY-NC 4.0 |
| CAMPR3 [18] | 4519 | CC BY-NC 4.0 |
| TumorHoPe [3] | 787 | CC BY 2.5 /3.0 |
| APD3[19] | 3072 | CC BY-NC 4.0 |
| SPdb [20] | 2512 | CC BY 2.0 |
| ParaPep [21] | 194 | CC BY [1] |
| CancerPPD [2] | 556 | CC BY-NC 4.0 |
| BrainPreps [5] | 92 | © Springer Nature [2] |
| Quorumpeps [22] | 257 | CC BY-NC 3.0 |
| YADAMP [15] | 2133 | © Elsevier [2] |
| LAMP2 [23] | 23 253 | CC BY [1] |
| Milkampdb [24] | 260 | CC BY [1] |
| DADP [12] | 2557 | © Oxford University Press [2] |
| AntiTbPdb [25] | 271 | CC BY 4.0 |
| PeptideDB [26] | 1903 | CC BY 4.0 |
| NeuroPrep [4] | 3875 | CC BY 4.0 |
| SATPdb [11] | 9664 | CC BY-NC 4.0 |
| BioDADPep [27] | 2543 | CC BY [1] |
| NeuroPedia [13] | 847 | © Oxford University Press [2] |
| DFBP [14] | 7058 | © Oxford University Press [2] |
| dbAMP [10] | 35 602 | CC BY-NC 4.0 |
| DRAMP [9] | 11 614 | CC BY 4.0 |
| AVPdb [1] | 2683 | CC BY 3.0 |
| Hemolytik [28] | 523 | CC BY-NC 3.0 |
| DBAASP [8] | 22 724 | CC BY-NC 4.0 |
| Uniprot [7] | 62 453 | CC BY 4.0 |

---

[1]The dataset authors reference a general license (e.g., Creative Commons) without specifying the exact version or associated terms.

[2]The dataset authors do not explicitly state the license governing the use of their data, and reuse must follow the specific terms set by the respective publishers or journals under standard academic publishing policies. Attribution through proper citation of the original sources is required for any use of these datasets. To ensure compliance with these conditions, we configure our dataset access pipeline to retrieve data directly through the official APIs or download interfaces provided by the original sources.

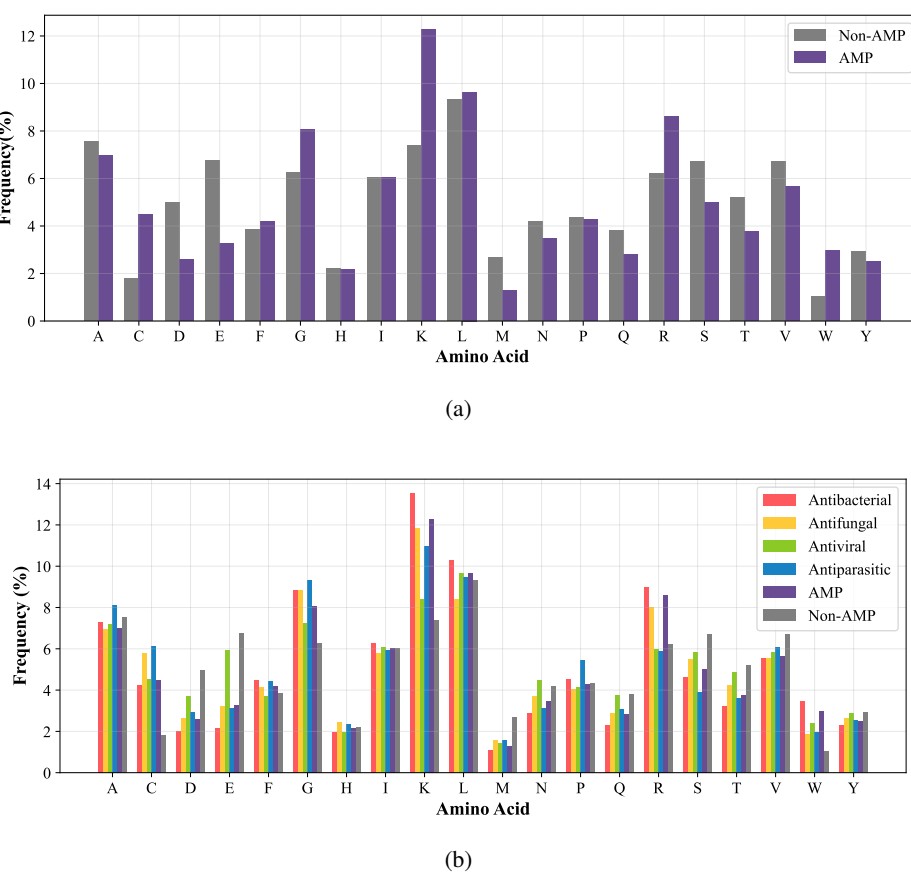

(a)

(b)

Figure 1: **Comparison of amino acid distributions in the ESCAPE dataset.** (a) Amino acid distributions for AMPs and Non-AMPs, with frequency differences reflecting variations between functional and non-functional peptides. (b) Normalized amino acid distributions with respect to each class for the multilabel clasification task. Overall, the dataset maintains a consistent aminoacid composition across categories.

## A.3   Statistical Analysis of the ESCAPE Dataset

We analyze the amino acid distribution across the five classes in the ESCAPE Database: antibacterial, antifungal, antiviral, antiparasitic, and antimicrobial peptides (AMP), along with the non antimicrobial peptides (Non-AMP). The frequencies vary across categories, reflecting differences in peptide counts. As an example, we compare the amino acid distributions between AMP and Non-AMP in Figure 1a. Since the Non-AMP category contains 2.85 times more peptides than AMP, its amino acid frequencies are higher. Figure 1b shows the normalized amino acid distributions across ESCAPE dataset classes and Non-AMP. Despite differences in the number of peptides per category, the relative frequencies remain largely consistent. This suggests that the dataset maintains a coherent overall amino acid composition across categories, with minimal variation. Moreover, it indicates that the underlying sequence composition remains stable, even across functionally distinct peptide groups.

## B   AMP Models on the ESCAPE Benchmark

### B.1   Implementation Details

We implement and evaluate all models mentioned in Section 3.4 to address the multilabel classification task. To ensure a comprehensive and representative benchmark, the evaluation includes a diverse set of model architectures: an attention-based LSTM [29], a random forest classifier [30], a graph neural network [31], and four Transformer-based models [32, 6, 33, 34], two of which leverage BERT

backbones [32, 6] and two that employ the vanilla Transformer architecture with physicochemical and sequence-related features [33, 34]. Moreover, AVP-IFT [34] also employs a contrastive learning module. Table 2 summarizes the training hyperparameters used for all models evaluated in this study, with the exception of amPEPpy [30]. For amPEPpy, which is based on a random forest classifier, we employ an ensemble of 160 bootstrap-aggregated decision trees and assess generalization performance using out-of-the-bag (OOB) estimation.

To address the multilabel classification task, we configure each model to output a binary vector of length five, where each dimension corresponds to one of the target antimicrobial classes: antibacterial, antiviral, antifungal, antiparasitic, and antimicrobial. We apply a sigmoid activation function to the final layer to produce independent probability estimates for each class. To adapt AVP-IFT [34] to the multilabel classification task, we changed the original binary similarity label in the contrastive loss to a continuous value that represents the fraction of similarity and dissimilarity across the five classes in the multilabel vector. We train each model separately on two distinct data folds and use a cross-validation setup to encourage generalization and reduce overfitting. During inference, we average the output logits from both trained instances before applying the sigmoid activation. This ensembling strategy treats both models as equal contributors and integrates their predictions into a single output.

Table 2: **Training hyperparameters for the implemented AMP deep learning models.** For each deep learning model trained in the ESCAPE Benchmark we show the training and model architecture hyperparameters.

| Hyperparameter | AMPlify [29] | AMP-BERT [32] | TranslmbAMP [6] | AMPs-Net [31] | PEP-Net [33] | AVP-IFT [34] |
|---|---|---|---|---|---|---|
| Max Length | 200 | 200 | 180 | — | 40 | 250 |
| Batch Size | 32 | 1 | 64 | 64 | 256 | 64 |
| Epochs | 70 | 15 | 256 | 300 | 100 | 100 |
| Learning Rate | $1 \cdot 10^{-3}$ | $1 \cdot 10^{-5}$ | $4 \cdot 10^{-2}$ | $5 \cdot 10^{-5}$ | $1 \cdot 10^{-4}$ | $1 \cdot 10^{-3}$ |
| Optimizer | Adam | Adam | Adam | Adamax | Adam | Adam |
| Dropout Rate | 0.5/0.2 | 0.0 | 0.2 | 0.2 | 0.5 | 0.5 |
| Hidden Dimension | 512 | 1024 | 512 | 256 | 1024 | 566 |
| Attention Heads | 32 | 16 | 12 | — | 4 | 2 |
| Activation | ReLU | GELU | Leaky ReLU | ReLU | ELU | ReLU |
| Transformer Layers | — | 30 | 12 | — | 1 | 1 |

## B.2 Statistical Significance of AMP Models in the ESCAPE Benchmark

We adopt a two-fold cross-validation strategy by training each model independently on two complementary folds of the dataset. For each trained instance, we evaluate performance on the corresponding test set using overall metrics, namely mean average precision (mAP) and F1 score, as well as class-wise scores for the five antimicrobial categories. We report these metrics separately for each fold to assess statistical consistency and variability across data partitions. To summarize model performance, we calculate the mean and standard deviation across the two folds, providing a reliable estimate of average predictive accuracy and performance variability. For this evaluation, we report the results with the 42 seed. Table 3 presents the F1-scores, while Table 4 reports the corresponding mAP values.

In Section 5.1, we report the performance of the ESCAPE baseline, defined as the ensemble of two independently trained models with logits averaged prior to the sigmoid activation. The ESCAPE Baseline ensemble using seed 42 achieves an overall F1 score of 69.5 and a mAP of 72.7, surpassing the mean performance of individual folds by 3.35 and 5.90 points, respectively. These results highlight the benefit of combining complementary representations learned from distinct training partitions, leading to improved robustness and predictive accuracy.

We analyze the relationship between model performance and the number of trainable parameters to examine the trade-off between predictive accuracy and architectural complexity. Figure 2 presents this comparison across all evaluated methods. Models with fewer parameters frequently achieve

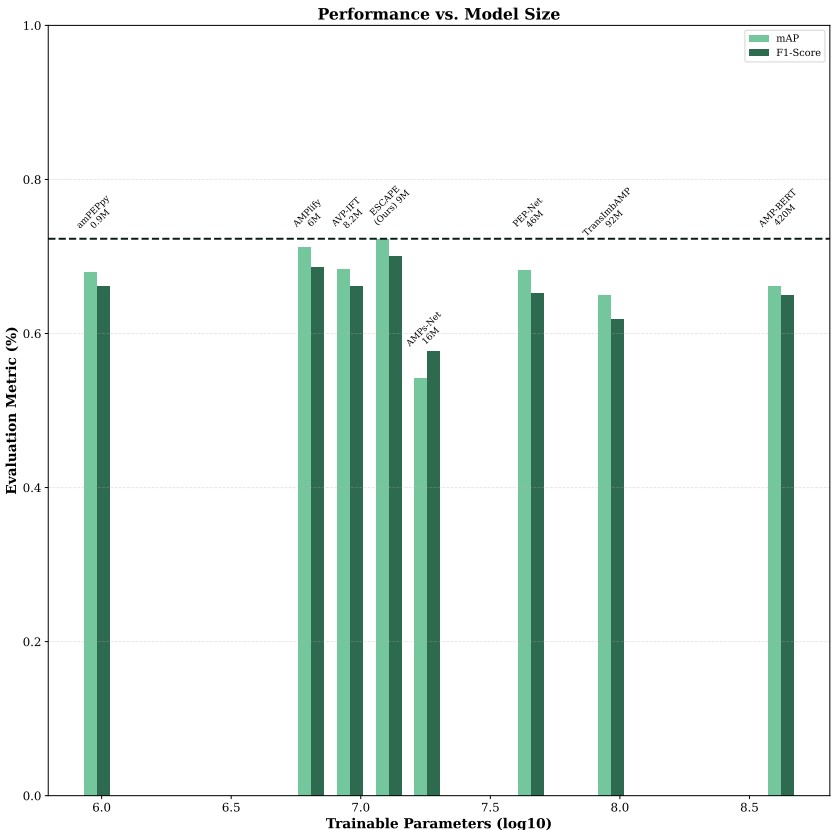

Figure 2: **Comparison of model performance and number of trainable parameters across all evaluated methods.** Since lighter models like the ESCAPE Baseline and AMPlify [29] show the best ensemble results in the test split and heavier models (e.g., BERT-based transformers [6] [32]) yield lower performance, we observe no consistent correlation between model size and predictive capability. Specifically, the ESCAPE Baseline achieves the best overall results with a fraction of the parameters used by large transformer models, suggesting that performance gains can be attained without increased model complexity.

Table 3: **Overall and Per-Class F1-Scores on the ESCAPE Benchmark without the logits ensemble strategy.** F1-scores for each model on the 5-class multilabel classification task in the ESCAPE Benchmark (%). We report values as mean ± standard deviation. These results correspond to the 42 random seed trained models.

| Method | F1-Score | Antibacterial | Antiviral | Antifungal | Antiparasitic | Antimicrobial |
|---|---|---|---|---|---|---|
| AMPs-Net [31] | $55.75 \pm 0.78$ | $77.9 \pm 0.57$ | $56.3 \pm 1.41$ | $53.65 \pm 0.77$ | $4.0 \pm 0.14$ | $80.65 \pm 1.48$ |
| TranslmbAMP [6] | $60.21 \pm 0.68$ | $86.03 \pm 0.03$ | $56.6 \pm 0.87$ | $52.05 \pm 0.76$ | $20.36 \pm 1.65$ | $85.99 \pm 0.13$ |
| AMP-BERT [32] | $63.49 \pm 0.54$ | $86.99 \pm 0.15$ | $58.56 \pm 0.21$ | $54.07 \pm 1.24$ | $20.37 \pm 2.31$ | $87.75 \pm 0.54$ |
| PEP-Net [33] | $62.86 \pm 1.89$ | $\mathbf{88.52 \pm 0.27}$ | $\mathbf{60.93 \pm 3.84}$ | $55.55 \pm 2.04$ | $19.46 \pm 2.63$ | $\mathbf{89.80 \pm 0.65}$ |
| amPEPpy [30] | $63.60 \pm 0.17$ | $86.71 \pm 0.14$ | $59.32 \pm 0.63$ | $55.61 \pm 1.75$ | $28.30 \pm 1.50$ | $88.08 \pm 0.17$ |
| AVP-IFT [34] | $61.32 \pm 0.60$ | $87.27 \pm 0.02$ | $59.83 \pm 0.21$ | $55.42 \pm 1.34$ | $15.74 \pm 3.71$ | $88.32 \pm 0.44$ |
| AMPlify [29] | $65.10 \pm 0.78$ | $87.61 \pm 0.14$ | $59.32 \pm 0.63$ | $55.55 \pm 1.37$ | $32.95 \pm 1.70$ | $88.04 \pm 0.52$ |
| **ESCAPE Baseline (Ours)** | $\mathbf{66.15 \pm 0.07}$ | $86.85 \pm 0.35$ | $58.88 \pm 1.06$ | $\mathbf{56.00 \pm 0.00}$ | $40.45 \pm 2.19$ | $88.55 \pm 0.21$ |

higher mAP and F1 scores, indicating that increased model size does not inherently translate to improved predictive capacity. However, when focusing on the amPEPpy, AMPlify, and ESCAPE models, a more consistent trend emerges: performance gains align with substantial increases in model size. For instance, achieving an improvement of less than 10% in mAP and F1 requires scaling from 0.9 million to over 6 million parameters. These findings underscore the importance of balancing model complexity with practical performance benefits when designing architectures for antimicrobial peptide classification.

Table 4: **Mean and Per-Class AP Results on the ESCAPE Benchmark without the logits ensemble strategy.** AP for each model on the 5-class multilabel classification task in the ESCAPE Benchmark (%). We report values as mean ± standard deviation. These results correspond to the 42 random seed trained models.

| Method | mAP | Antibacterial | Antiviral | Antifungal | Antiparasitic | Antimicrobial |
|---|---|---|---|---|---|---|
| AMPs-Net [31] | $51.50 \pm 0.85$ | $80.30 \pm 1.27$ | $47.30 \pm 1.98$ | $49.50 \pm 1.77$ | $4.70 \pm 0.14$ | $79.15 \pm 0.92$ |
| TransImbAMP [6] | $62.57 \pm 0.11$ | $91.98 \pm 0.08$ | $62.21 \pm 0.25$ | $52.48 \pm 0.01$ | $13.27 \pm 0.56$ | $92.90 \pm 0.14$ |
| AMP-BERT [32] | $64.44 \pm 0.36$ | $90.58 \pm 0.65$ | $63.52 \pm 0.05$ | $56.35 \pm 0.52$ | $16.91 \pm 0.13$ | $91.41 \pm 0.06$ |
| amPEPpy [30] | $65.12 \pm 0.17$ | $93.30 \pm 0.15$ | $63.24 \pm 0.68$ | $\mathbf{57.83 \pm 0.39}$ | $16.38 \pm 0.32$ | $94.84 \pm 0.07$ |
| PEP-Net [33] | $65.43 \pm 1.58$ | $\mathbf{94.54 \pm 0.17}$ | $\mathbf{67.79 \pm 4.44}$ | $57.61 \pm 1.53$ | $12.14 \pm 2.23$ | $\mathbf{95.48 \pm 0.16}$ |
| AVP-IFT [34] | $62.08 \pm 1.13$ | $91.30 \pm 1.98$ | $63.50 \pm 1.79$ | $55.78 \pm 1.28$ | $7.38 \pm 2.04$ | $92.46 \pm 1.13$ |
| AMPlify [29] | $64.25 \pm 0.01$ | $91.14 \pm 1.57$ | $61.02 \pm 0.86$ | $55.01 \pm 0.04$ | $21.55 \pm 4.29$ | $92.52 \pm 1.84$ |
| **ESCAPE Baseline (Ours)** | $\mathbf{66.80 \pm 0.42}$ | $91.55 \pm 0.35$ | $63.25 \pm 0.35$ | $56.70 \pm 1.56$ | $\mathbf{28.50 \pm 0.42}$ | $94.05 \pm 0.07$ |

## C   Sensitivity of the model to predicted 3D protein structures

To further evaluate the influence of structural inputs on the ESCAPE Baseline, we conducted an ablation study in which we replaced experimental 3D crystal structures with AlphaFold-predicted counterparts. Experimental structures were available for only 2.086 peptides from UniProt (2.5% of ESCAPE), distributed as 846 in Fold 1, 825 in Fold 2, and 415 in Test. In our experiment, the experimental structures in Fold 1 and Fold 2 (1.671 peptides) were replaced with their AlphaFold predictions, while the Test set remained unchanged. Results indicate that using only predicted structures reduces performance relative to experimental data (Table 5), with absolute drops of 1.5% (mAP) and 1.9% (F1). These findings confirm that experimental crystal structures provide superior inputs for the structure module, but also show that predicted structures remain a viable alternative when experimental data are unavailable. This sensitivity highlights the dependence of the ESCAPE Baseline on the quality of structural representations, suggesting that future improvements in structure prediction methods, such as AlphaFold [35] and RosettaFold [36], may directly enhance classification performance.

Table 5: **Ablation experiments for the ESCAPE Baseline with respect to 3D predicted protein structures.** These results correspond to the 42 random seed trained model.

| Training Data | mAP | F1-Score |
|---|---|---|
| Only generated structures | 71.2 | 67.5 |
| Experimental + generated structures | 72.7 | 69.4 |