# OpenReview forum: "A Standardized Benchmark for Multilabel Antimicrobial Peptide Classification"
_NeurIPS.cc/2025/Datasets_and_Benchmarks_Track — NeurIPS 2025 Datasets and Benchmarks Track poster_

### Official Review · Reviewer_F5DT · 2025-06-03

**Rating:** 5
**Confidence:** 3

**Summary:**

This paper presents ESCAPE, a large, standardized dataset comprising over 80,000 antimicrobial peptide (AMP) sequences annotated with multiple functional labels, including antibacterial, antifungal, antiviral, and antiparasitic activities. It resolves the fragmentation and inconsistency issues found in prior AMP datasets. Furthermore, the authors introduce a transformer-based model that integrates both sequence and structural information to predict peptide functionalities with high accuracy. Their method achieves state-of-the-art performance and offers a robust framework to accelerate AI-driven AMP discovery, thereby supporting global efforts against antimicrobial resistance.

**Dataset Code Accessibility:**

Yes

**Ethical Considerations:**

No, there are no or only very minor ethics concerns

**Limitations Weaknesses:**

The evaluation algorithms employed are relatively outdated, with only one method published in 2024 and the others predating 2023. It is recommended to incorporate more state-of-the-art (SOTA) methods for the assessment. Furthermore, the adapted prior methods tailored for this specific task have generally exhibited decreased performance.

**Strengths Contributions:**

1. The authors introduce ESCAPE, the first standardized benchmark for multi-label antimicrobial peptide classification. This benchmark is crucial for enabling fair and consistent comparisons in this specific task.

2. The paper is well written and easy to follow. Furthermore, the code has been made public to enhance its reproducibility.

3. Extensive experiments highlight the advantages of ESCAPE in effectively evaluating multi-label antimicrobial peptide classification models. Notably, the proposed method achieves up to a 2.15% improvement compared to methods tailored for this specific task.

---

> ### Author Rebuttal · Authors · 2025-07-31
>
> We thank the reviewer F5DT for their thoughtful and constructive feedback regarding the implemented algorithms. We appreciate the emphasis on including more state-of-the-art (SOTA) methods, and we agree that such comparisons are crucial for strengthening the relevance and impact of our benchmark. Below, we provide a discussion of our baseline selection, the limitations encountered with several recent models, and the integration of an additional recent method into the evaluation.
>
> **Selection of Evaluation Algorithms**
>
> In selecting baselines for our evaluation, we prioritized methods that are widely used and have publicly available code. While several of the comparative methods included in the manuscript (AMPlify, TransImbAMP, amPEPpy, and AMPs-Net) were published prior to 2023, they remain among the most relevant and representative approaches in the field. For instance, AMPlify (2022) achieved the second-best performance among all compared models in both mean average precision (mAP) and F1-score, outperforming PEP-Net (2024).
>
> **Limitations in Recent Methods**
>
> To provide a broader and more up-to-date comparison, we explored the implementation of several recently proposed methods, including StackAMP (2024) [1], AMP-RNNpro (2024) [2], dsAMPGAN (2024) [3], AMPpred-DLFF (2024) [4], and StackPIP (2025) [5]. StackAMP combines predictions from multiple models trained on different peptide features to improve AMP classification. AMP-RNNpro introduces a Recurrent Neural Network architecture that leverages eight encoding strategies, including compositional, grouped, autocorrelation, and pseudo-amino acid features. dsAMPGAN integrates CNN, Attention, and BiLSTM layers with transfer learning to perform AMP classification and function prediction. AMPpred-DLFF employs ESM-2 (an amino acid sequence-based language model) embeddings and graph attention networks to capture spatial residue relationships, while also incorporating convolutional modules for comprehensive feature extraction. StackPIP proposes a stacking ensemble strategy using 12 machine learning algorithms and diverse peptide descriptors to enhance peptide interaction prediction.
>
> Despite the relevance of these recent methods, we were unable to reproduce or evaluate them within our benchmark due to the absence of publicly available code, complete training pipelines, or sufficiently detailed documentation. This limited our ability to conduct a fair and rigorous comparison under the unified framework established in the ESCAPE Benchmark. By contrast, ESCAPE was developed with reproducibility and transparency as core principles. We provide open access to the ESCAPE Dataset, along with training and evaluation code for the ESCAPE Baseline. Nonetheless, we will include these recent approaches in the related work section to acknowledge their contributions and ensure a more comprehensive overview of the current landscape in AMP prediction.
>
> **Adaptation of a Recent SOTA Method (AVP-IFT, 2024)**
>
> In light of the reviewer’s suggestion to incorporate more recent methods, we took the initiative to broaden our evaluation by adapting AVP-IFT (2024) [6] to the ESCAPE Benchmark. Originally developed for antiviral peptide classification, AVP-IFT consists of two separate branches that are pretrained independently. The first branch employs a contrastive learning approach to capture discriminative representations of peptide sequences, while the second branch employs a transformer network enhanced with various peptide features. Following pretraining, the representations from both branches are combined and passed to a classification module. To evaluate it fairly on our benchmark, we adapted the method to handle multi-class antimicrobial peptide classification, thereby extending its scope beyond its original focus on antiviral sequences.
>
> When evaluated on the ESCAPE Benchmark, AVP-IFT achieved an average F1-score of 66.1% and a mean average precision (mAP) of 68.3%. These results, like all those reported in our benchmark, correspond to a single run with the same random seed (42) to ensure consistency and comparability across models. While AVP-IFT’s performance is competitive, it remains inferior to that of our proposed method. Notably, in the antiviral class, AVP-IFT achieved the highest performance among all evaluated models, with an F1-score of 64.8% and a mAP of 70.8%, likely reflecting its original design, which was specifically tailored for antiviral peptide classification.
>
> Although certain models may outperform ours in specific categories, ESCAPE Baseline demonstrates more consistent results across the full task, reflecting a higher degree of generalization in the multi-class AMP setting and its ability to learn representations for each category. This highlights the value of ESCAPE not only as a benchmark for assessing per-class accuracy but also as a robust framework for evaluating overall performance. We will include AVP-IFT in the final version of the manuscript to acknowledge its contributions and to provide a more complete and up-to-date comparison within the benchmark.
>
> **References**
>
> [1] Tasmin Karim, Md. Shazzad Hossain Shaon, Md. Mamun Ali, Kawsar Ahmed, Francis M. Bui, and Li Chen. StackAMP: Stacking-based ensemble classifier for antimicrobial peptide identification. *IEEE Transactions on Artificial Intelligence*, 5(11):5666–5675, 2024.
>
> [2] Md. Shazzad Hossain Shaon, Tasmin Karim, Md. Fahim Sultan, Md. Mamun Ali, Kawsar Ahmed, Md. Zahid Hasan, Ahmed Moustafa, Francis M. Bui, and Fahad Ahmed Al-Zahrani. AMP-RNNpro: a two-stage approach for identification of antimicrobials using probabilistic features. *Scientific Reports*, 14(1):12892, 2024.
>
> [3] Min Zhao, Yu Zhang, Maolin Wang, and Luyan Z. Ma. dsAMP and dsAMPGAN: Deep Learning Networks for Antimicrobial Peptides Recognition and Generation. *Antibiotics (Basel)*, 13(10):948, 2024.
>
> [4] Yu Chen, Xingpeng Jiang, and Weizhong Zhao. AMPpred-DLFF: prediction of AMPs based on deep learning and multi-view features fusion. In *Proceedings of the 2024 IEEE International Conference on Bioinformatics and Biomedicine (BIBM)*, pages 891–896, 2024.
>
> [5] Lantian Yao, Feng Wang, Peilin Xie, Jiahui Guan, Zhihao Zhao, Xuxin He, Xingchen Liu, Ying-Chih Chiang, and Tzong-Yi Lee. StackPIP: An Effective Computational Framework for Accurate and Balanced Identification of Proinflammatory Peptides. *Journal of Chemical Information and Modeling*, 65(14):7777–7788, 2025.
>
> [6] Jiahui Guan, Lantian Yao, Peilin Xie, Chia-Ru Chung, Yixian Huang, Ying-Chih Chiang, and Tzong-Yi Lee. A two-stage computational framework for identifying antiviral peptides and their functional types based on contrastive learning and multi-feature fusion strategy. *Briefings in Bioinformatics*, 25(3):bbae208, 2024.

---

### Official Review · Reviewer_3Rfb · 2025-06-29

**Rating:** 4
**Confidence:** 3

**Summary:**

This manuscript presents a benchmark dataset and classification framework for identifying antimicrobial peptides (AMPs). The goal is to train models capable of distinguishing AMP sequences from non-AMPs, potentially aiding in the discovery of novel therapeutic peptides. The authors compiled over 80,000 AMP sequences from 27 databases and standardized their annotations into five categories to support more consistent training and evaluation. In addition, they developed a baseline model and compared its performance against several state-of-the-art (SOTA) methods. Their results suggest that sequence information contributes more significantly to classification performance than structural features.

**Dataset Code Accessibility:**

Yes

**Dataset Code Comments:**

Data and code are accessible. But I have not tested it.

**Ethical Considerations:**

No, there are no or only very minor ethics concerns

**Final Justification:**

the authors have addressed my concerns. I increase my rating to 4.

**Limitations Weaknesses:**

1. The validity of the ground truth labels is unclear. It would be important to specify whether the AMP annotations are experimentally validated or computationally predicted, as this impacts the reliability of the training data.

2. Negative samples are drawn only from Uniprot peptide sequences. This could limit the diversity of negative examples. Including randomly generated peptides or peptides from other functionally distinct families might make the task more robust?

3. Figure 2b suggests a strong length bias between AMP and non-AMP sequences. If AMPs are consistently shorter, models may rely primarily on sequence length for classification. The authors should control for this confounder in training and evaluation.

4. The inputs and training conditions of the benchmarked SOTA methods are not well-documented. It would be helpful to categorize the methods based on input types (e.g., sequence-only, structure-aware) and clarify whether these models were retrained on the new benchmark dataset or used as-is. This distinction is critical for a fair comparison.

5. The conclusions would be more compelling if ablation studies (e.g., testing performance with and without structural features) were also applied to the SOTA methods, not just the baseline.

**Strengths Contributions:**

1. The dataset is large and aggregated from 27 sources, representing a considerable effort in data collection.

2. The authors have standardized annotations across sources, which helps unify heterogeneous data.

3. A baseline method is provided for benchmarking, and results show improved performance over selected SOTA models.

4. The study provides insights into the relative contribution of sequence versus structural features for AMP classification.

---

> ### Author Rebuttal · Authors · 2025-07-31
>
> We sincerely thank Reviewer 3Rfb for the detailed feedback and constructive suggestions. We appreciate the acknowledgment of our dataset’s scale, standardized annotations, and benchmarking efforts. Below, we address the concerns raised regarding ground-truth validation, negative sample diversity, sequence length bias, and documentation of state-of-the-art (SOTA) methods. We have revised the manuscript accordingly and provided clarifications.
>
>
> **Ground Truth Labels Validity**
>
> We fully agree that the quality and reliability of ground truth labels directly impact model performance. To clarify, all AMP annotations in ESCAPE are based on experimental validation, not computational predictions. Positive examples include peptides with confirmed antimicrobial activity, as demonstrated through *in vitro* or *in vivo* assays, obtaining Minimum Inhibitory Concentration (MIC) values using standardized protocols (e.g., CLSI[1]). Negative examples are peptides with experimentally verified non-antimicrobial functions (e.g., hormones, signaling peptides) and no reported antimicrobial activity. We explicitly excluded peptides with *in silico* predictions or low-confidence annotations to ensure high-quality, biologically reliable ground truth. We will revise the manuscript to make this clarification more explicit and enhance transparency regarding the validity of the ground truth labels.
>
>
> **Negative Samples Diversity**
>
> We appreciate the reviewer’s comment regarding the diversity of negative samples and recognize that the composition of the negative class is crucial for ensuring robust and meaningful model evaluation. In response, we clarify that the negative samples in ESCAPE Dataset were carefully curated to prioritize biological reliability and experimental validation, rather than relying on randomly generated or purely synthetic sequences. All negative peptides are experimentally validated for functions unrelated to antimicrobial activity (e.g., hormones, signaling molecules, structural peptides). These were sourced from both UniProt and curated databases with explicit non-AMP annotations (e.g., BioDADPep, ParaPep). Our strategy combined (1) exclusion filtering in UniProt, removing peptides linked to antimicrobial-related terms like “toxic,” “secretory,” or “defensive”, and (2) inclusion of peptides from datasets with functions far from antimicrobial mechanisms, such as neuropeptides and tumor-targeting peptides. We intentionally avoided synthetic or randomly generated negatives, which may introduce unrealistic biases and inflate performance. Instead, the negative examples in ESCAPE Dataset reflect real biological diversity and challenge models to learn meaningful discriminative features. We will revise the final version of the paper to clearly describe this strategy and emphasize the biological validity of our negative class.
>
>
> **Correlation Between Sequence Length and Classification Performance**
>
> Regarding the potential length bias highlighted in Figure 2b, we acknowledge that AMPs are generally shorter than non-AMPs, a characteristic rooted in their biological function. To evaluate whether this difference affects model behavior, we performed a correlation analysis between peptide length and model predictions, measured as the L1-norm between the model’s output probabilities and the annotations for both AMP and non-AMP sequences. We computed Pearson correlation coefficients separately for positive and negative classes. For AMPs, we obtained a correlation of 0.273 (p < 0.01), and for non-AMPs, a correlation of –0.253 (p < 0.01), both of which are statistically significant. However, the magnitude of the correlation in both cases is low, indicating only a weak relationship between sequence length and model predictions.
>
> To further investigate the potential influence of sequence length on model performance, we computed the F1-score for each individual sample in the test set and conducted a linear regression using sequence length as the independent variable. The resulting Pearson correlation coefficient (r) was 0.26, indicating a weak positive linear relationship. Therefore, while sequence length may contribute marginally to the classification decision, it does not appear to be a dominant or heavily relied-upon feature in the model’s predictions. Nevertheless, we acknowledge this weak correlation as a potential source of bias and will include it in our discussion as one of the limitations of the current benchmark.
>
>
> **Documentation and Reproducibility of Benchmarking Methods**
>
> We thank the reviewer for emphasizing the importance of transparent benchmarking. In ESCAPE Benchmark, all selected SOTA models were retrained from scratch on our dataset using the same data splits, preprocessing steps, and evaluation metrics to ensure fair and reproducible comparisons. Each model included in the benchmark has a publicly available and reproducible implementation, which allowed us to reliably adapt it to our multilabel classification task. No method was evaluated “as-is”, all models were retrained and optimized on ESCAPE Dataset, regardless of their original domain or objective. Additionally, to improve clarity, we will revise the manuscript to explicitly categorize the benchmarked models according to their input types. These include models that use only sequence information, such as AMPlify and AMP-BERT; models that incorporate structural information, such as ESCAPE Baseline; models based on physicochemical features, such as AMPs-Net and amPEPpy; and models that rely on pretrained embeddings, such as PEP-Net.
>
>
> **Justification for Model-Specific Structural Ablations**
>
> We appreciate the suggestion to extend ablation studies to benchmarked SOTA models. However, we clarify that none of the other compared methods include a structural information module; they rely solely on sequence information (e.g., AMPlify, AMP-BERT), physicochemical features (e.g., AMPs-Net, amPEPpy), or pretrained sequence embeddings (e.g., PEP-Net). As such, they are not directly comparable in terms of structural ablation. In contrast, ESCAPE Baseline is the only model that integrates both sequence and 3D structural data. To fairly assess the impact of structural information, we conducted detailed ablation experiments within our baseline model (Section 5.2), isolating the contributions of sequence, structure, and their combination via cross-attention. Importantly, all models were trained from scratch under the same unified experimental protocol on the ESCAPE benchmark, ensuring a fair and consistent comparison across methods.
>
> **References**
>
> [1] CLSI. Methods for Dilution Antimicrobial Susceptibility Tests for Bacteria That Grow Aerobically. 12th ed. CLSI standard M07. *Clinical and Laboratory Standards Institute*; 2024.

---

### Official Review · Reviewer_P24Q · 2025-07-01

**Rating:** 5
**Confidence:** 2

**Summary:**

The work introduces ESCAPE, a multilabel benchmark that unifies more than 80 000 peptide sequences from 27 public repositories under a consistent antimicrobial activity hierarchy. Beyond curating the dataset, the authors establish a three-fold evaluation protocol and reimplement six representative classifiers, thereby delivering the reproducible, large-scale comparison for multilabel AMP prediction. They further propose a dual-branch transformer baseline that fuses sequence embeddings with 3D distance-matrix representations via bidirectional cross-attention; this model raises m-AP by 2.3 percentage points and F1 by 1 point over the strongest prior system on the new benchmark .

**Dataset Code Accessibility:**

Yes

**Dataset Code Comments:**

The authors commit to releasing all data, code, and benchmark splits, which will facilitate reproducibility and foster broader community adoption.

**Ethical Considerations:**

Yes, there are ethics concerns that require attention by the authors

**Final Justification:**

I have reviewed the authors’ response, which has addressed most of my questions. I will therefore maintain my positive score.

**Limitations Weaknesses:**

More discussions or evidence could be considered. For example, how sensitive is the proposed model to errors in the predicted structures from AlphaFold or RosettaFold? Also, can the authors provide confidence intervals for each class to clarify whether the reported 2.3% improvement in m-AP is statistically meaningful across different data splits and random seeds?

Regarding the class imbalance highlighted in the dataset. The antiparasitic class has only 417 entries, with 130 being unique. Such underrepresentation might lead to poor model generalization for minority classes, as seen in the lower performance metrics for antiparasitic peptides despite the baseline's improvements.

The paper's baseline model uses a transformer architecture with cross-attention between sequence and structural data. However, the ablation study shows that structural data alone underperforms compared to sequence data. This raises questions about the added value of structural information, especially when predictive inaccuracies are considered. The marginal gains from incorporating structure might not justify the computational overhead.

While the paper discusses broader impacts and the need for clinical validation, it doesn't address potential biases from integrating multiple databases. Differences in experimental protocols across sources could introduce noise, affecting model training and performance consistency.

**Strengths Contributions:**

The study addresses a critical gap in AMP informatics by standardizing previously heterogeneous databases and introducing clearly defined train/validation/test splits.  Its multilabel annotation framework reflects the complex, real-world spectrum of antimicrobial activity, allowing models to predict activity across multiple pathogen types simultaneously, rather than being constrained to binary classifications. The semi-automated data curation process demonstrates a high level of methodological rigor and transparency. Also, the authors evaluate six representative architectures under controlled conditions and provide ablation studies that disentangle the individual contributions of sequence and structural features, thereby strengthening the validity of their findings. The authors commit to releasing all data, code, and benchmark splits, which will facilitate reproducibility and foster broader community adoption.

---

> ### Author Rebuttal · Authors · 2025-07-31
>
> We sincerely thank the reviewer for the thoughtful and constructive feedback. We are grateful for the recognition of our efforts in unifying a large-scale, multilabel AMP dataset, establishing a reproducible evaluation framework, and proposing a novel cross-modal transformer baseline. We address concerns on structure, significance, imbalance, trade-offs, and bias.
>
> **Sensitivity of the Model to Structural Prediction**
>
> We agree that further evidence regarding the structural representations used in the ESCAPE Baseline would strengthen the work, and we appreciate the reviewer’s comments concerning this aspect. For the structure module of the ESCAPE Baseline, our goal was to utilize the largest amount of experimental data, specifically the 3D crystal structures of the peptides. However, for the 27 compiled databases, only a subset of 2086 peptides from UniProt provided these experimental structures. This subset represents only 2.5% of the total structures in the ESCAPE Database and is distributed as follows: 846 in Fold 1, 825 in Fold 2, and 415 in Test.
>
> Following the reviewer’s suggestion, we conducted a new ablation experiment in which we replaced the experimental crystal structures in Fold 1 and Fold 2 with their AlphaFold-predicted versions (a total of 1,671 structures). The Test set was left unchanged to ensure a fair comparison of the results with those from the ESCAPE Baseline.
>
> |          **Training Data**          | **mAP** | **F1-Score** |
> |:-----------------------------------:|:-------:|:------------:|
> |      Only Generated Structures      |   71.2  |     67.5     |
> | Experimental + Generated Structures |   72.7  |     69.4     |
>
> We observe a decrease in performance when replacing the experimental structures with their AlphaFold-predicted counterparts, with an absolute drop of 1.5% in mAP and 1.9% in F1-score. This experiment demonstrates that, although experimental structures provide better input for the structure module, predicted structures remain a viable alternative when experimental data is unavailable. Moreover, this result highlights an important avenue for future improvement as the performance of the ESCAPE Baseline is directly influenced by the quality of structural inputs. With structure prediction models such as AlphaFold and RosettaFold continuing to advance, their increasing accuracy is likely to yield more informative structural representations, which can enhance the classification performance of our method. We will incorporate this structural dependency into the discussion on the current limitations of our benchmark.
>
> **Assessing Statistical Significance of mAP Improvement**
>
> We agree with the reviewer on the importance of evaluating the statistical significance of the 2.3% mAP improvement using confidence intervals. In response, we ran all methods with three random seeds (42, 1665, and 8914) for consistency. Here we present the tables with the confidence intervals:
>
>
> |     Method Name    |   mAP   | Antibacterial AP | Antiviral AP | Antifungal AP | Antiparasitic AP | Antimicrobial AP |
> |:----------------------:|:-----------:|:--------------------:|:----------------:|:-----------------:|:--------------------:|:--------------------:|
> |        AMPs-Net        | 54.6 ± 0.86 |      82.5 ± 0.72     |    51.2 ± 0.88   |    53.1 ± 0.84    |      5.3 ± 0.67     |      82.1 ± 0.80     |
> |       TransImbAMP      | 64.9 ± 1.11 |      92.5 ± 1.23     |    65.0 ± 1.63   |    56.3 ± 0.96    |      16.7 ± 0.86     |      94.0 ± 0.90     |
> |        AMP-BERT        | 66,9 ± 1.17 |      92.3 ± 0.59     |    65.9 ± 1.84   |    61.5 ± 2.28    |      21.4 ± 2.61     |      93.6 ± 1.25     |
> |         amPEPpy        | 68.5 ± 0.48 |      93.9 ± 0.24     |    67.7 ± 0.28   |    62.2 ± 0.27    |      23.8 ± 1.61     |      95.2 ± 0.05     |
> |         PEP-Net        | 68.4 ± 0.53 |      95.2 ± 0.21     |    61.2 ± 0.67   |    72.6 ± 0.78    |      16.2 ± 0.84     |      96.7 ± 0.26     |
> |         AMPlify        | 70.3 ± 0.87 |      94.0 ± 0.19     |    66.1 ± 5.56   |    68.3 ± 4.27    |      27.7 ± 1.33     |      95.3 ± 0.31     |
> | ESCAPE Baseline (Ours) | 72.1 ± 0.60 |      94.2 ± 0.21     |    69.8 ± 0.46   |    63.4 ± 0.74    |      37.6 ± 2.87     |      95.6 ± 0.04     |
>
> |     Method Name    |    F1   | Antibacterial F1 | Antiviral F1 | Antifungal F1 | Antiparasitic F1 | Antimicrobial F1 |
> |:----------------------:|:-----------:|:--------------------:|:----------------:|:-----------------:|:--------------------:|:--------------------:|
> |        AMPs-Net        | 57.7 ± 0.70 |      78.9 ± 0.77     |    59.2 ± 0.79   |    61.1 ± 0.51    |      5.9 ± 0.71     |      83.5 ± 0.79     |
> |       TransImbAMP      | 62.0 ± 0.70 |      87.1 ± 0.96     |    59.2 ± 0.50   |    54.7 ± 0.51    |      21.8 ± 0.81     |      87.2 ± 0.75     |
> |        AMP-BERT        | 64.7 ± 0.64 |      89.3 ± 0.27     |    63.0 ± 0.95   |    60.2 ± 0.26    |      20.6 ± 3.52     |      90.5 ± 0.22     |
> |         amPEPpy        | 66.5 ± 0.37 |      87.6 ± 0.07     |    61.6 ± 2.02   |    60.4 ± 1.90    |      34.7 ± 0.98     |      90.9 ± 3.78     |
> |         PEP-Net        | 65.5 ± 0.61 |      89.5 ± 0.10     |    58.1 ± 0.78   |    65.2 ± 0.55    |      22.8 ± 0.61     |      91.2 ± 0.15     |
> |         AMPlify        | 68.5 ± 0.77 |      88.8 ± 0.26     |    60.0 ± 1.05   |    65.0 ± 1.57    |      40.9 ± 2.48     |      90.0 ± 0.30     |
> | ESCAPE Baseline (Ours) | 69.8 ± 0.43 |      88.8 ± 0,34     |    64.4 ± 0.88   |    61.0 ± 0.75    |      44.8 ± 0.50     |      90.0 ± 0.32     |
>
> The results confirm that ESCAPE Baseline outperformed all other methods in both mAP and F1-Score. The ranking remained mostly unchanged, with only a slight swap between amPEPpy and PEP-Net for mAP. Importantly, there is no overlap in the confidence intervals between ESCAPE Baseline and AMPlify, indicating that the improvement is statistically significant.
>
> **Class Imbalance in the Dataset**
>
> We acknowledge the significant class imbalance, particularly in the antiparasitic category, which affects generalization and is reflected in lower performance. This limitation arises from the biological and experimental landscape, not curation choices. ESCAPE Database includes only experimentally validated peptides from curated repositories and peer-reviewed literature. Despite integrating 27 databases, only 417 antiparasitic examples are currently available with confirmed in vitro or in vivo activity. Expanding this class would require low-confidence or in silico data, which we excluded to preserve benchmark reliability.
>
> Reflecting this scarcity is crucial to keep the benchmark close to the real scenario, as it mirrors the actual experimental bias in AMP discovery pipelines. Furthermore, the lower representation of antiparasitic peptides highlights an open research challenge in developing robust methods against extreme class imbalance. Notably, our method continues to achieve incremental gains in this category, demonstrating its ability to learn from limited data. ESCAPE thus offers a high-confidence, biologically grounded benchmark under realistic data constraints for advancing antiparasitic AMP prediction.
>
> **Integration of the Structural Module**
>
> Regarding the marginal gain from incorporating the structural module, we appreciate the reviewer’s point and thank them for raising this issue. While the structural module alone underperforms compared to the sequence module (47.7% mAP vs. 69.4%, and 46.9% F1-Score vs. 67.6%), the ablation studies presented in Table 3 clearly demonstrate the value of combining both modalities. When integrated via cross-attention, the model achieves 72.7% mAP and 69.5% F1, outperforming both individual modules.
>
> This shows that structural information adds non-redundant cues not captured by sequence data, and the cross-attention mechanism effectively aligns these modalities to improve performance. While the standalone structural module is weaker due to predicted 3D structure noise, its integration helps disambiguate functional patterns.
>
> Although adding the structural module increases parameters, Supplementary Figure 2 shows that our model’s parameter count remains lower than most state-of-the-art methods and slightly higher than AMPlify. The sequence-only version also remains competitive with leading methods, offering a lightweight alternative with strong performance.
>
> Furthermore, the flexibility of our baseline model is a key advantage. It offers two versions: the full version with structural data, which outperforms all methods, and the sequence-only version, which is more parameter-efficient yet still highly competitive. This versatility makes ESCAPE Baseline adaptable to varying resource constraints and performance needs
>
> **Potential Biases from Integrating Multiple Databases**
>
> We thank the reviewer for highlighting the potential biases arising from integrating multiple data sources. To mitigate this, the ESCAPE dataset was designed with strict inclusion criteria based solely on experimentally validated data.
> For positive samples, we included only antimicrobial peptides with experimentally determined Minimum Inhibitory Concentration (MIC) values, obtained under standardized protocols (e.g., CLSI guidelines [1]). These measurements ensure reproducibility and consistency across laboratories. Additionally, databases such as DBAASP, APD3, and DRAMP only accept AMP annotations that have undergone peer-reviewed experimental validation, thereby reinforcing data reliability.
> For negative samples, we selected peptides with experimentally confirmed non-antimicrobial functions (e.g., hormones, signaling, or structural peptides), ensuring they have no reported antimicrobial activity. This reduces the risk of including false negatives.
>
> **References**
>
> [1] CLSI. Methods for Dilution Antimicrobial Susceptibility Tests for Bacteria That Grow Aerobically. 12th ed. CLSI standard M07. *Clinical and Laboratory Standards Institute*; 2024.

---

### Official Review · Reviewer_eiL4 · 2025-07-02

**Rating:** 4
**Confidence:** 3

**Summary:**

This paper introduces a dataset of over 80,000 peptides sourced from 27 different repositories, designed to support multilabel classification of antimicrobial peptides (AMPs). In addition to the dataset, the authors propose a dual-branch transformer-based model that integrates sequence and structural information. They benchmark its performance against several baselines. The authors argue that their approach leads to improvements in metrics such as mean Average Precision (mAP) and F1 score.

**Additional Feedback:**

The authors use a transformer architecture based on a dual-branch design to jointly encode the sequence and structural modalities of peptides. Their comparisons show that BERT-based approaches perform poorly in this domain, and they underscore the limitations of large language models when applied outside of natural language. Structured state space models have recently shown promising results on DNA data, often matching or outperforming transformers while using significantly fewer resources. Have the authors explored this?

**Dataset Code Accessibility:**

Yes

**Ethical Considerations:**

No, there are no or only very minor ethics concerns

**Final Justification:**

I appreciate the authors’ response to my comments, particularly the effort to re-run all methods with different random seeds. With the latest updates, I believe the score of 4 is more appropriate.

**Limitations Weaknesses:**

Despite the contributions, the paper suffers from several inconsistencies in its presentation, reporting, and analysis that undercut the strength of its empirical claims:

- Figure 2a uses grey color for bars, matching the color for the Non-AMP label, which can be misleading.
- The caption for subfigure 2(a) mentions four functional classes in the ESCAPE dataset, but 2(b) and 2(c) show six labels, including antimicrobial and Non-AMP. Line 136 mentions five classes. This should be made consistent across the text, figures, and supplementary materials.
- Figure 2c: The caption refers to training, validation, and test splits, but the subfigure shows Fold1, Fold2, and Test. Please clarify.
- Line 172: The authors state that the combination of antibacterial and antifungal peptides includes 6,652 sequences, but based on Figure 2a, it appears to be 4,960 (3718 + 1104 + 107 + 31).
- Line 282: The paper claims that the ESCAPE Baseline improves AP for the antiparasitic class by 31% over the second-best method. However, Table 2 shows only a 9.5% improvement.
- In the abstract, the authors mention up to a 2.15% improvement in mAP, but this value is not shown or discussed elsewhere. Table 2 suggests a mAP gain of 1.6% compared to the second-best method.
- Table 1: The highest performance in the antimicrobial class is achieved by PEP-Net (91.1%), not ESCAPE (89.7%), which is bolded.

Overall, the reported improvements in mAP are promising, especially in underperforming classes, but the paper does not include standard deviations or confidence intervals across multiple runs. Given the modest overall gains and some per-class fluctuations, reporting variance would strengthen the empirical claims and help assess whether the improvements are statistically meaningful. The inconsistencies in how results are reported are concerning. Addressing these issues would strengthen both the technical and communicative aspects of the work.

**Strengths Contributions:**

The paper addresses an important area at the intersection of peptide research and machine learning. The scale and diversity of the dataset are impressive, and the effort to unify data from 27 repositories adds clear value to the community. The task formulation as a multilabel AMP classification problem is sensible. The paper provides empirical comparisons to a range of baselines using common metrics like mAP and F1 score. Though the overall improvement in the F1 score (+0.5) appears incremental rather than substantial, the AP results show a clearer and more consistent improvement, especially in the antiparasitic class (+9.5).

---

> ### Author Rebuttal · Authors · 2025-07-31
>
> We thank Reviewer eiL4 for the detailed and constructive feedback. The comments highlight several important issues regarding class definitions, figure labeling, and metric reporting, which are essential for improving the clarity and consistency of our manuscript.
>
> **Figure 2a - Color Misinterpretation**
>
> Regarding Figure 2a, we note that the bars share the same grey color as the Non-AMP label, which may cause confusion. We will change the color of the bars for the camera-ready version to avoid this confusion and improve visual clarity.
>
> **Class Definitions and Labeling Consistency**
>
> Regarding the perceived inconsistency in the number of classes presented across the text and figures, we clarify that the ESCAPE Dataset defines four main functional classes: antibacterial, antifungal, antiparasitic, and antiviral. The fifth class, labeled as antimicrobial, is not a new or separate class but rather an aggregate label used to denote peptides that belong to any of the four functional categories above. This allows us to distinguish between peptides with known antimicrobial activity (i.e., those that fall under at least one of the four functional classes) and those without such annotation. Thus, peptides not belonging to any of the four classes are considered non-AMPs and serve as our negative examples. This organization enables us to model both fine-grained classification across functional types and binary discrimination between AMPs and Non-AMPs
>
> Lastly, it is important to clarify that the term non-AMP, as used in our manuscript and in Figure 2, does not denote an additional class. Rather, it refers to samples that do not belong to any of the five defined classes and are considered negative examples. Hence, we will revise the figure captions and manuscript text to clarify these distinctions and ensure consistency across all sections.
>
> **Figure 2c - Dataset Splits**
>
> In Figure 2c, the caption refers to training, validation, and test splits, while the figure itself uses the terms Fold1, Fold2, and Test, which is an incompatibility between the caption and the labels. Since we use a 2-fold cross-validation strategy, the dataset is divided into Fold1, Fold2, and the Test set, as shown in figure 2c. We will modify the caption accordingly to improve consistency.
>
> **Misreported Number of Sequences**
>
> We also appreciate the reviewer’s attention to a misreported number: the manuscript states that the combination of antibacterial and antifungal peptides includes 6,652 sequences, but the correct value, as shown in Figure 2a, is 4,960. This discrepancy was based on a miscalculation of the number of antifungal peptides only, and we will revise the manuscript to reflect the accurate total.
>
> **Improvement in Average Precision (AP) for Antiparasitic Class**
>
> The reviewer also noted a mismatch in the reported improvement in average precision (AP) for the antiparasitic class. The manuscript currently states a 31% improvement on the antiparasitic class, which refers to the relative increase, computed as [(40.1 − 30.6)/30.6] × 100% = 31%. However, we understand that this may be misinterpreted as an absolute percentage point difference (which would be 9.5%), and we will revise the manuscript to clearly indicate that this is a relative improvement compared to the second-best method.
>
> **Abstract’s mAP Improvement and Table 1 Formatting Issue**
>
> A similar situation applies to the abstract, where we reported a 2.15% improvement in mAP. This value contains a typo, and the correct relative improvement is 2.25%, based on the comparison between 72.7 and 71.1 as shown in Table 2. We will correct the value and explicitly describe it as a relative improvement. Additionally, we acknowledge the formatting issue in Table 1, where ESCAPE Baseline is incorrectly bolded as the best-performing method in the antimicrobial class. The highest performance is achieved by PEP-Net (91.1%), and we will correct this in the camera-ready version.
>
> **Variance Metrics and Statistical Significance**
>
> We fully agree that reporting variance metrics is essential to reinforce the robustness and reproducibility of our empirical findings. Following the reviewer's suggestion, we undertook the additional effort of re-running all methods included in the benchmark two more times using different random seeds. Specifically, we systematically employed the same three seed values (42, 1665, 8914) across all methods to ensure consistency and comparability. This involved re-executing every baseline and our proposed model under identical experimental conditions, which we considered a valuable step to further strengthen the validity of our empirical claims.
>
> Here we present the results including mean and standard deviation across the three independent runs. We report these metrics for both mAP and F1 scores, at both the global level and disaggregated per class. This update will be incorporated into the final version of the paper for the camera-ready submission.
>
> |     Method Name    |   mAP   | Antibacterial AP | Antiviral AP | Antifungal AP | Antiparasitic AP | Antimicrobial AP |
> |:----------------------:|:-----------:|:--------------------:|:----------------:|:-----------------:|:--------------------:|:--------------------:|
> |        AMPs-Net        | 54.6 ± 0.86 |      82.5 ± 0.72     |    51.2 ± 0.88   |    53.1 ± 0.84    |      5.3 ± 0.67     |      82.1 ± 0.80     |
> |       TransImbAMP      | 64.9 ± 1.11 |      92.5 ± 1.23     |    65.0 ± 1.63   |    56.3 ± 0.96    |      16.7 ± 0.86     |      94.0 ± 0.90     |
> |        AMP-BERT        | 66,9 ± 1.17 |      92.3 ± 0.59     |    65.9 ± 1.84   |    61.5 ± 2.28    |      21.4 ± 2.61     |      93.6 ± 1.25     |
> |         amPEPpy        | 68.5 ± 0.48 |      93.9 ± 0.24     |    67.7 ± 0.28   |    62.2 ± 0.27    |      23.8 ± 1.61     |      95.2 ± 0.05     |
> |         PEP-Net        | 68.4 ± 0.53 |      95.2 ± 0.21     |    61.2 ± 0.67   |    72.6 ± 0.78    |      16.2 ± 0.84     |      96.7 ± 0.26     |
> |         AMPlify        | 70.3 ± 0.87 |      94.0 ± 0.19     |    66.1 ± 5.56   |    68.3 ± 4.27    |      27.7 ± 1.33     |      95.3 ± 0.31     |
> | ESCAPE Baseline (Ours) | 72.1 ± 0.60 |      94.2 ± 0.21     |    69.8 ± 0.46   |    63.4 ± 0.74    |      37.6 ± 2.87     |      95.6 ± 0.04     |
>
> |     Method Name    |    F1   | Antibacterial F1 | Antiviral F1 | Antifungal F1 | Antiparasitic F1 | Antimicrobial F1 |
> |:----------------------:|:-----------:|:--------------------:|:----------------:|:-----------------:|:--------------------:|:--------------------:|
> |        AMPs-Net        | 57.7 ± 0.70 |      78.9 ± 0.77     |    59.2 ± 0.79   |    61.1 ± 0.51    |      5.9 ± 0.71     |      83.5 ± 0.79     |
> |       TransImbAMP      | 62.0 ± 0.70 |      87.1 ± 0.96     |    59.2 ± 0.50   |    54.7 ± 0.51    |      21.8 ± 0.81     |      87.2 ± 0.75     |
> |        AMP-BERT        | 64.7 ± 0.64 |      89.3 ± 0.27     |    63.0 ± 0.95   |    60.2 ± 0.26    |      20.6 ± 3.52     |      90.5 ± 0.22     |
> |         amPEPpy        | 66.5 ± 0.37 |      87.6 ± 0.07     |    61.6 ± 2.02   |    60.4 ± 1.90    |      34.7 ± 0.98     |      90.9 ± 3.78     |
> |         PEP-Net        | 65.5 ± 0.61 |      89.5 ± 0.10     |    58.1 ± 0.78   |    65.2 ± 0.55    |      22.8 ± 0.61     |      91.2 ± 0.15     |
> |         AMPlify        | 68.5 ± 0.77 |      88.8 ± 0.26     |    60.0 ± 1.05   |    65.0 ± 1.57    |      40.9 ± 2.48     |      90.0 ± 0.30     |
> | ESCAPE Baseline (Ours) | 69.8 ± 0.43 |      88.8 ± 0,34     |    64.4 ± 0.88   |    61.0 ± 0.75    |      44.8 ± 0.50     |      90.0 ± 0.32     |
>
> The findings show that, on average, the ESCAPE Baseline achieved superior performance across both mAP and F1-Score compared to all other methods. Under this more rigorous evaluation, the overall ranking of methods remained largely stable: the F1-Score rankings were unchanged, while for mAP, the only difference was an exchange in positions between amPEPpy and PEP-Net. Notably, ESCAPE Baseline consistently obtains the top position in both metrics.
>
> **Structured State Space Models - Future Work**
>
> We thank the reviewer once again for their constructive feedback and for taking the time to provide this additional suggestion. Regarding the comment on structured state space models, we would like to acknowledge that this is a very insightful direction that we had not originally considered during the development of our work.
>
> Prompted by the reviewer’s suggestion, we conducted a brief literature review and found that structured state space models have indeed shown promising results in biological sequence modeling tasks such as those involving DNA, sometimes matching or outperforming transformer-based models, while being computationally more efficient [1]. This opens a compelling line of research that could be highly relevant in the context of peptide function prediction as well.
>
> We believe this feedback highlights an exciting opportunity for future work. Beyond the scope of the current study, we hope that the unified, experimentally validated dataset introduced in this paper, along with the rigorous evaluation protocol and baseline comparisons, provides a strong foundation upon which future efforts can build. Our goal for this benchmark is to foster further exploration in the community, enabling fair comparisons across architectures and accelerating progress in peptide-based research.
>
> **References**
>
> [1] Popov, M., Kallala, A., Ramesh, A., Hennouni, N., Khaitan, S., Gentry, R., and Cohen, A.-S: Leveraging state space models in long range genomics. *LMRL Workshop at ICLR*, 2025.

---

> > ### Comment · Reviewer_eiL4 · 2025-08-05
> >
> > I appreciate the authors’ response to my comments, particularly the effort to re-run all methods with different random seeds. They have addressed all of my concerns, and I will be updating my score accordingly.

---

> > > ### Author Response · Authors · 2025-08-06
> > > **Comment to Reviewer eiL4:**
> > >
> > > We sincerely thank Reviewer eiL4 for the thorough and constructive feedback.
> > > The comments were instrumental in clarifying several aspects of the work and in guiding additional experiments that strengthened and supported the results.
> > > We greatly appreciate the time and effort dedicated to this review, and all the discussed points and corrections will be incorporated into the camera-ready version of the manuscript.

---

### Note · Authors · 2025-08-15

We thank the AC and reviewers for their constructive feedback. The reviewers recognized the value of ESCAPE as: (i) The first large-scale, experimentally validated, multilabel benchmark for antimicrobial peptide classification, integrating over 80,000 sequences from 27 curated repositories under a unified hierarchical annotation framework (eiL4, P24Q, 3Rfb, F5DT); (ii) A transparent, reproducible evaluation protocol with clear splits for fair, consistent benchmarking of diverse methods. (P24Q, 3Rfb, F5DT); (iii) Comprehensive reimplementation and retraining of representative baselines under fair conditions, facilitating robust comparisons (eiL4, P24Q, 3Rfb); and (iv) A novel dual-branch transformer baseline integrating sequence and structural modalities via cross-attention, achieving consistent improvements across classes while maintaining adaptability to different resource constraints (P24Q, F5DT).

The rebuttal period allowed us to clarify class definitions, correct minor reporting inconsistencies, and incorporate additional experiments including structure-sensitivity tests, variance reporting, correlation analysis for potential biases –no significant correlation with sequence length– and integration of a recent SOTA model. All discussed clarifications, and corrections will be incorporated in the Camera-Ready version, ensuring that the final manuscript fully reflects the constructive dialogue with the reviewers.

Reviewer eiL4 - We resolved class definition and figure-label inconsistencies, and re-ran all methods with three random seeds to provide mean ± std metrics, improving clarity and statistical robustness.

Reviewer P24Q - We performed a structural sensitivity analysis, reported confidence intervals confirming the significance of mAP gains, and contextualized antiparasitic scarcity as a biologically driven limitation, enriching the discussion on applicability and constraints.

Reviewer 3Rfb - We documented experimental validation criteria for all labels, detailed the curation of diverse negative samples, quantified and contextualized sequence length as no bias, and categorized benchmarked methods by input type, enhancing transparency and fairness.

Reviewer F5DT - We adapted and integrated the recent AVP-IFT (2024) model into our benchmark, reaffirming the generalization capacity of our baseline. Additionally, we will include in the related work five other recent but non-reproducible methods to provide a more complete overview of the field.

---

### Decision · Program_Chairs · 2025-09-18

**Decision:**

Accept (poster)

**Comment:**

The paper presents a huge dataset of antimicrobial peptides by collecting and standardizing data from various sources. Baselines and a better-performing custom transformer model are also considered.

The main concerns of the reviewers targeted the presentation, potential label biases, and smaller items, all of which got addressed in the responses. The reviewers acknowledged the considerable effort in data collection, additional insights of the analysis, and potential impact of the benchmark. They provided detailed reviews and, after submitting very mixed initial ratings, came to a common (borderline) accept recommendation. Alas, the majority did not provide more justification for their final ratings, even upon request.